EMBO
Molecular Medicine

# A novel platform for attenuating immune hyperactivity using EXO-CD24 in COVID-19 and beyond

Shiran Shapira[1,2], Marina Ben Shimon[1], Mori Hay-Levi[1], Gil Shenberg[1], Guy Choshen[3], Lian Bannon[4] (iD), Michael Tepper[1], Dina Kazanov[1], Jonathan Seni[1,2], Shahar Lev-Ari[1,2] (iD), Michael Peer[5], Dimitrios Boubas[6], Justin Stebbing[7] (iD), Sotirios Tsiodras[6] & Nadir Arber[1,2,*] (iD)

## Abstract

A small but significant proportion of COVID-19 patients develop life-threatening cytokine storm. We have developed a new anti-inflammatory drug, EXO-CD24, a combination of an immune checkpoint (CD24) and a delivery platform (exosomes). CD24 inhibits the NF-kB pathway and the production of cytokines/chemokines. EXO-CD24 discriminates damage-from pathogen-associated molecular patterns (DAMPs and PAMPs) therefore does not interfere with viral clearance. EXO-CD24 was produced and purified from CD24-expressing 293-TREx™ cells. Exosomes displaying murine CD24 (mCD24) were also created. EXO-CD24/mCD24 were characterized and examined, for safety and efficacy, *in vitro* and *in vivo*. In a phase Ib/IIa study, 35 patients with moderate–high severity COVID-19 were recruited and given escalating doses, $10^8$–$10^{10}$, of EXO-CD24 by inhalation, QD, for 5 days. No adverse events related to the drug were observed up to 443–575 days. EXO-CD24 effectively reduced inflammatory markers and cytokine/chemokine, although randomized studies are required. EXO-CD24 may be a treatment strategy to suppress the hyper-inflammatory response in the lungs of COVID-19 patients and further serve as a therapeutic platform for other pulmonary and systemic diseases characterized by cytokine storm.

**Keywords** CD24; COVID-19; cytokine storm; exosomes; EXO-CD24
**Subject Categories** Immunology; Microbiology, Virology & Host Pathogen Interaction; Pharmacology & Drug Discovery
**EMBO Mol Med (2022) e15997**

## Introduction

Severe acute respiratory syndrome (SARS) coronavirus 2 (SARS-CoV-2) is a newly discovered member of the family of coronaviruses. Most infected individuals are asymptomatic or suffer from mild symptoms including fever, fatigue, and dry cough. In a minority of cases, after several days, clinical deterioration may occur rapidly over a short period of 12–24 h leading to an acute respiratory distress syndrome (Chen *et al*, 2020).

Therapy is required in many of those infected with comorbid diseases (Bhatraju *et al*, 2020; Goossens *et al*, 2020; Verity *et al*, 2020). These patients are prone to develop acute respiratory distress syndrome (ARDS), pulmonary insufficiency, a need for ventilation and even death (Hu *et al*, 2021). All these severe circumstances are characterized by the cytokine storm (Channappanavar & Perlman, 2017; Fajgenbaum & June, 2020).

COVID-19 pathology presents a particular therapeutic challenge as it stems from both rampant viral infection and immune over-reactivity (Channappanavar & Perlman, 2017; Garcia, 2020).

Cluster of differentiation (CD) CD24 is a small, heavily glycosylated, membrane-anchored protein which functions as an immune checkpoint regulator (Bradley, 2019; Qiu *et al*, 2021).

CD24 allows immune discrimination between damage-associated molecular patterns (DAMPs) released from damaged or dying cells, and pathogen-associated molecular patterns (PAMPs) derived from pathogens such as bacteria and viruses. The binding of CD24 to DAMPs prevents them from binding to pattern recognition receptors, such as Toll-like receptors (TLRs), and inhibition of DAMP-activation of the nuclear factor-κB (NF-κB) pathway, a key signaling pathway driving production of cytokines and chemokines (Liu *et al*, 2009; Barkal *et al*, 2019). Another distinct class of pattern recognition receptors are Siglecs. CD24 binds both DAMPs and

1  The Health Promotion Center and Integrated Cancer Prevention Center, Tel Aviv, Israel
2  Department of Molecular Genetics and Biochemistry, Sackler Faculty of Medicine, Tel Aviv University, Tel Aviv, Israel
3  Department of Internal Medicine H, Tel Aviv Sourasky Medical Center, Tel Aviv, Israel
4  Department of Internal Medicine F, Tel Aviv Sourasky Medical Center, Tel Aviv, Israel
5  Department of Thoracic Surgery, Tel Aviv Sourasky Medical Center, Tel Aviv, Israel
6  4th Department of Internal Medicine, Attikon University Hospital, National and Kapodistrian University of Athens Medical School, Athens, Greece
7  Department of Surgery and Cancer, Imperial College, London, UK
   *Corresponding author. Tel: +972 3 6974968/3716; Fax: +972 3 6974867; E-mail: nadira@tlvmc.gov.il

Siglec-10 resulting in activation of the Siglec-10 signaling pathway (Chen et al, 2009; Barkal et al, 2019). This pathway negatively regulates the activity of NF-κB, through immunoreceptor tyrosine-based inhibition motif (ITIM) domains associated with SHP-1 (SRC homology 2 (SH2)-domain-containing protein tyrosine phosphatase-1). It is also supported by the identification of a recombinant fusion protein composed of the extracellular domain of CD24 linked to a human immunoglobulin G1 Fc domain as a potential immune checkpoint inhibitor with anti-inflammatory activity (Toubai et al, 2017; Tian et al, 2018). While CD24 dampens DAMPs initiated immune activation, it does not affect PAMPs immune recognition, and hence does not interfere with viral clearance.

Exosomes are intraluminal vesicles which play a role in intercellular communication (Colombo et al, 2014; Yáñez-Mó, 2015). Exosomes increase stability and enhance the bioavailability of bioactive compounds (Amreddy et al, 2018; Guo & Jiang, 2020; Elliott & He, 2021). They are in ongoing clinical research as carriers of therapeutic agents against cancer, cardiovascular diseases, diabetesc, graft-versus-host, neurological, and orthopedic diseases (Wolfers et al, 2001; Dai et al, 2008; Le Blanc et al, 2008; Heldring et al, 2015; Newton et al, 2017). A recent study found that exosomes delivered via nebulizer can help repair pulmonary fibrosis-induced lung injury (Dinh et al, 2020).

We have developed CD24-enriched exosomes, named EXO-CD24, as a targeted therapy for hyperimmune activation in the context of COVID-19. CD24, which dampens cytokines and chemokines production while not interfering with pathogen clearance, is of particular interest as a therapeutic agent for virus-induced hyper-inflammation and ARDS.

EXO-CD24, and its mouse homolog EXO-mCD24, is a new precision nanotechnology that can target and prevent the cytokine storm in the lungs. Herein, the safety and efficacy of EXO-CD24 is shown in vitro, in vivo and in a phase Ib/IIa clinical study.

# Results

### EXO-CD24 production and characterization

We previously described achieving tightly regulated, tetracycline inducible, overexpression of human CD24 following stable transfection of cloned CD24 into a HEK-293-derived cell-line. We utilized this CD24-stably transfected cell line for the purification of exosomes displaying high levels of CD24. Briefly, CD24 expression was induced by tetracycline for 72 h. Cell growth medium was collected, and CD24-exosomes were isolated using ExoQuick-CG. Isolated exosomes are herein referred to as EXO-CD24. Induced cells demonstrated high expression of EXO-CD24 using both Western blots analysis (Fig 1A) and flow cytometry (Fig 1B). The expression of CD24 in purified exosomes was detected by exosome-based ELISA (Fig 1C) and Western blots which confirmed that CD24 display only on exosomes isolated from cells induced by tetracycline to overexpress CD24 (Fig 1D). The expression of CD24 was also studied using flow cytometry. To allow for size approximation, fluorescent-stained beads of known sizes (100, 200, and 500 nm) were co-analyzed with the EXO-CD24 sample. As expected for known exosomal size, EXO-CD24 were predominantly found in the 100–200 nm size range. In addition,

flow cytometry analysis confirmed the copresence of CD24 and exosomal marker CD81 (Fig 1E). EXO-CD24 morphology was studied using cryo-electron microscopy (cryo-EM). The size and morphology of the exosomes with lipid bilayers and vesicular internal structures are clearly visible (Fig 1F). The presence of other exosomal biomarkers such as CD63, CD9, CD81, and HSP70 was detected by several methods (exosome-based ELISA, WB, and flow cytometry) and verified that the isolated population is exosomes.

### In vitro studies

#### EXO-CD24 inhibits PMA-induced cytokine/chemokine secretion in a human monocyte cell line

The effect of EXO-CD24 on pro-inflammatory cytokines and chemokines secretion was studied in an in vitro model of human monocytes using the cell line U937. Cells were stimulated with phorbol 12-myristate 13-acetate (PMA), a widely used agent for monocyte/macrophage immune activation. Untreated monocytes grew in suspension showing their known morphological characteristics of small round shaped cells (Fig 2A). PMA-treated cells resulted in their well-characterized phenotype of cell adherence and cell cycle arrest followed by differentiation (Fig 2A). As expected, in comparison with controls, EXO-CD24-treated cells secreted reduced levels of various cytokines and chemokines including IL-1β, RANTES, CD40, IL-1α, IL-6, MCP-1, and MIP-3a/CCL20 (Fig 2B).

To confirm the efficacy and specificity of EXO-CD24 treatment, the exact experiment was repeated, but with the addition of monoclonal antibodies against CD24 that were added simultaneously with EXO-CD24. The effect of the EXO-CD24 was partially blocked by these antibodies.

### In vivo studies

#### EXO-mCD24 dose toxicity study in mice

The safety of EXO-mCD24 administration was examined in a repeated-dose toxicity study in mice. For this purpose, exosomes presenting the murine homolog of CD24 (mCD24) were developed, by transiently induced high mCD24 expression in NIH3T3 murine cell line. A dose of either $5 \times 10^8$ or $1 \times 10^9$ EXO-mCD24 was administered by inhalation, once daily, for 5 days. Animals were either sacrificed on day 6 or followed for an additional week. Saline, the carrier for the exosomes, was used as vehicle. Administration method (inhalation) and duration (number of days) were designed to mimic the suggested therapeutic use in humans. No adverse effects or differences were observed between control and treated groups in behavior, food and water consumption, body weight, organ weight at the end of the study, nor in hematology, blood chemistry, and urine analyses (Fig 3A–E). In addition, histological evaluation of organs (brain, lung, heart, liver, kidneys, spleen, thymus, and thyroid) from five vehicle-treated and eight animals treated with $1 \times 10^9$ EXO-mCD24 showed no effects on organ histology.

#### EXO-mCD24 reduce cytokine release and lung inflammatory reaction in an ARDS model

Next, we investigated the efficacy of mCD24-exosomes in LPS-induced ARDS mouse model.

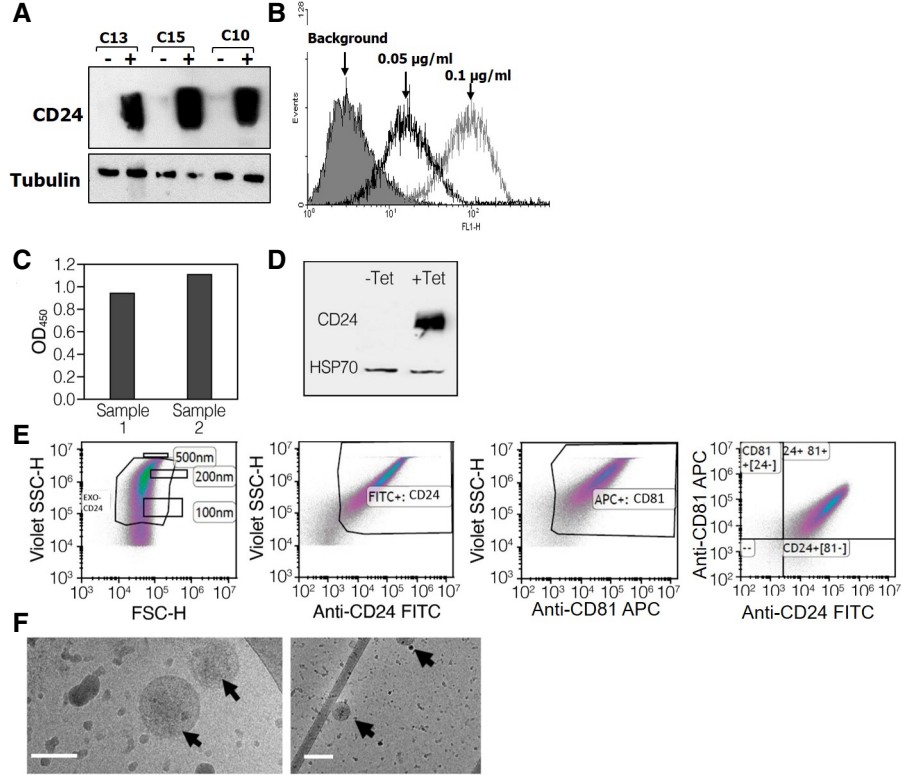

**Figure 1. Characterization of CD24-displaying exosomes (EXO-CD24).**

A Representative clones screened for CD24 expression under regulation of tetracycline. The cells were exposed to 1 µg/ml tetracycline for 48 h. Twenty microgram from each sample were subjected to Western blot analysis for CD24 and analyzed with anti-CD24 SWA11 mAb. The membrane was reprobed with anti-tubulin antibody to assess the uniformity of sample loading.

B The effect of tetracycline concentration on CD24 expression levels was examined by flow cytometry.

C Exosome-based ELISA was used to measure expression of CD24 in the purified exosomes.

D Western blot analysis of EXO-CD24 exosomes. Tetracycline (Tet) was used to induce CD24 expression in the cell line from which exosomes are derived. HSP70 was used as an exosomal marker.

E Flow cytometry analysis of EXO-CD24. Left panel: Cells were gated based on cell size (forward scatter [FSC] versus side scatter [SSC]); Beads of known sizes of 100, 200 and 500 nm were used for size reference; Middle-left panel: EXO-CD24 labeled with anti-CD24 FITC antibody and gated to the exosome population; Middle-right panel: EXO-CD24 labeled with anti-CD81 APC antibody and gated by to the exosome population; Right panel: EXO-CD24 double labeled with anti-CD24 FITC and anti-CD81 APC antibodies and gated by to the exosome population.

F Cryo-TEM images of isolated CD24-exosomes. On a number of different devices, many experiments were conducted and representative images are shown. Arrows point to double-membraned vesicles (exosomes). Scale bars are 100 nm (left panel) and 200 nm (right panel).

Exosomes presenting the murine homolog of CD24 (mCD24) were developed, by transiently induced high HSA expression (Fig 4A–C) in Expi293 embryonic cell line.

The animals were challenged by intratracheal introduction of LPS, followed by mCD24 exosome treatment, at $1 \times 10^8$ or $1 \times 10^9$ EXO-mCD24, once daily for 3 days and were then sacrificed (Fig 5A–C; $n = 10$ per group). Naïve mice (not challenged by LPS) served as overall control ($n = 5$) and saline-treated mice used as vehicle control treatment ($n = 10$). As expected, histology confirm normal healthy lungs in the naïve animals, while the lungs in all LPS-challenged groups had a multifocal coalescing inflammatory reaction, composed predominantly of neutrophils. The inflammatory infiltrates were mainly perivascular but also observed around the middle sized and small bronchiole (Fig 5). Animals treated with saline or low-dose EXO-mCD24 ($1 \times 10^8$) had moderate-to-severe lung injury with a score of $4.7 \pm 1.1$ and $4.6 \pm 0.8$, respectively. Animals in the high-dose treatment group ($1 \times 10^9$ EXO-mCD24)

presented a moderate lung injury, with a score of $4.0 \pm 0.8$. A marked reduction in cytokine and chemokine levels (IL-6, IL-12, TNF-α, IFN-γ, and KC/CXCL1) was observed, in animals treated with $1 \times 10^9$ EXO-mCD24 in the sera (Fig 5, upper panel) and in the BAL (Fig 5, lower panel), most of them in a dose-depended manner. These results suggest that treatment with CD24-exosomes may curtail inflammation.

### Inhaled EXO-mCD24 increase survival in a mouse sepsis model

EXO-mCD24 treatment was examined in a cecum ligation and puncture (CLP)-induced sepsis model in mice. The animals underwent the CLP procedure followed by vehicle- or EXO-mCD24-treatment, at $1 \times 10^9 / 1 \times 10^{10} / 1 \times 10^{11}$ mCD24-exosomes/mouse, by inhalation at 1-, 8-, and 24-h post-surgery. Analysis of serum cytokine levels did not show differences between the vehicle and treatment groups. However, the control group had higher mortality compared with the group treated with $1 \times 10^{10}$ EXO-mCD24 (Log Rank:

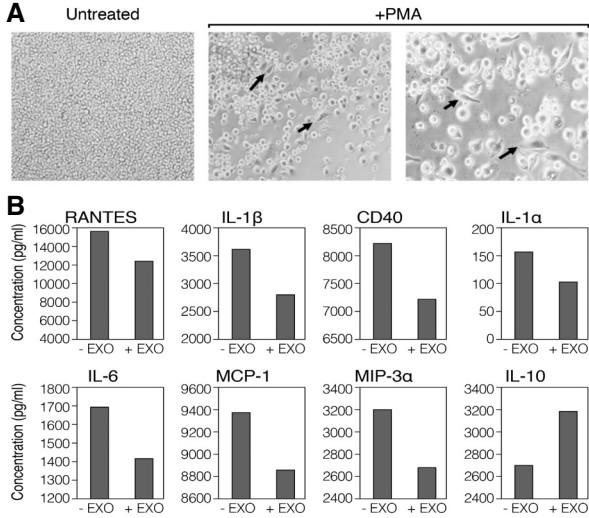

**Figure 2.** EXO-CD24 reduces *in vitro* expression of cytokines and chemokines.

A  PMA stimulation of monocyte cell line U937 show change in cell morphology and adherence. Untreated (left) and stimulated cells with PMA for 72 h (right). The arrows point to U937 macrophage-like cells (scale bar ×10).

B  U937 cells were stimulated with PMA and incubated with (+EXO) or without (−EXO) EXO-CD24 treatment. Incubation medium was then collected and analyzed for cytokines/chemokines levels using a "Multi-plex" array. Data shown are the average of duplicates from a single experiment.

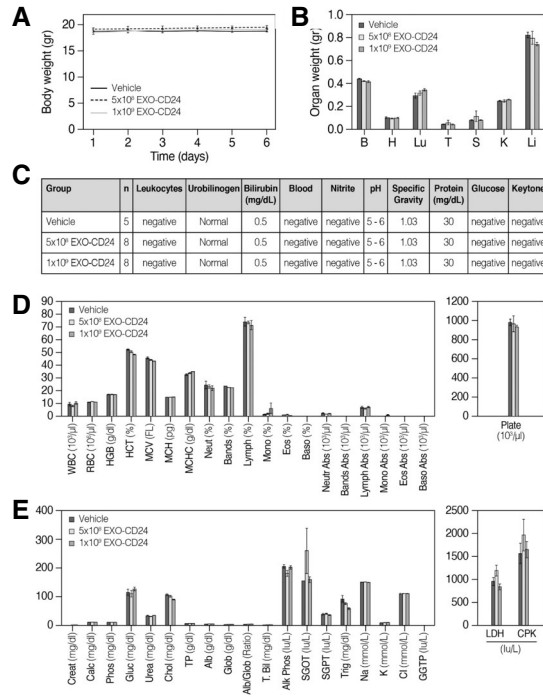

**Figure 3.** EXO-CD24 repeated-dose toxicity study in mice, 7-day follow-up.

A dose of either $5 \times 10^8$ and $1 \times 10^9$ exosomes/mouse ($n = 8$ mice/treated group in the main group, $n = 3$ mice/group in the follow-up groups) was administered by inhalation once daily, for 5 days. Saline, the carrier for the exosomes, was used as vehicle. Animals were followed up during the 5 treatment days and for 7 additional days before being sacrificed.

A  Daily body weights (gram). Body weight was measured daily during the first week and three times a week during the second week. Data are average ± SEM.

B  Organ weight following a full 5-day treatment course; B: brain, H: heart, Lu: lungs, T: thymus, S: spleen, K: kidneys, Li: liver. Data are average ± SEM.

C  Urine analysis. The urine was collected and analyzed and no abnormalities were found.

D  Hematology. Mice were anesthetized and blood was taken for full hematology tests. Data are average ± SEM.

E  Blood Chemistry. Mice were anesthetized and blood was taken for clinical chemistry tests. Data are average ± SEM.

Data information: Data are expressed as mean ± SEM; vehicle group $n = 5$ mice; $n = 8$ mice per treatment group. Raw values and statistical analyses are provided in Appendix Supplementary Methods.

---

$P < 0.001$) or $1 \times 10^{11}$ EXO-mCD24 (Log Rank: $P = 0.001$; Fig 6A). Cox regression showed longer survival in the $1 \times 10^{10}$ EXO-mCD24 group (hazard ratio = 0.069; 95% CI, 0.016–0.292) and in the $1 \times 10^{11}$ exosomes/mouse group (hazard ratio = 0.155; 95% CI, 0.043–0.555) compared with the control group. The mortality rate and survival were comparable in the $1 \times 10^9$ EXO-mCD24 and the control group (hazard ratio = 0.684; 95% CI, 0.255–1.838).

### Pharmacokinetic study

The pharmacokinetic and pharmacodynamics parameters had been evaluated, following a single inhalation of EXO-CD24. It was detected in the BALF (Fig 6B) and even in the bloodstream (Fig 6C), without accumulation in any organ.

### *Phase Ib/IIa clinical study*

Thirty-five patients were enrolled, between September 26[th], 2020, and February 13[th], 2021, in four dose escalation groups (Fig 7A).

Group 1: five patients that were enrolled one by one with a waiting time of at least 2 weeks between patients. They received EXO-CD24 at a concentration of $1 \times 10^8$ particles per 2 ml saline solution. Safety findings for each individual patient of the first group were reported to the hospital's IRB, and following approval by the IMOH, the next patient group was recruited. Subsequent groups were enrolled following a review of the safety data of the preceding group by the IMOH. Group 2: A dose escalation group of five patients received EXO-CD24 at a concentration of $5 \times 10^8$ particles per 2 ml saline solution. Following IMOH approval, the third group was recruited. Group 3: 20 patients received $1 \times 10^9$ exosomes. The

drug seemed to be very safe with possible efficacy. The IMOH granted recruitment of Group 4: five patients that received $1 \times 10^{10}$ exosomes.

Participants were free to withdraw from participation in the study at any time. All patients, except for one, completed the full treatment regimen of five inhalations and completed the study as planned.

The overall eligible study population was mostly comprised of men (65.7%). The mean ± SD participant age at baseline was 57.5 ± 11.46 (range 33–77) years. The mean time from COVID-19 diagnosis until the first treatment was 9.5 days. The mean, upon enrollment, blood oxygen saturation (SpO$_2$), and respiratory rate were 90.7% (range 90–94) and 27.6 (range 23–30) breaths/min, respectively. Average CRP prior to treatment was 127.0 ± 14.7

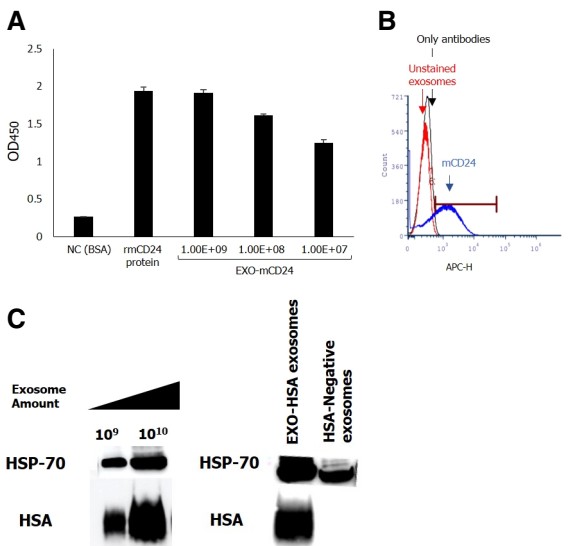

**Figure 4. Characterization of mCD24-displaying exosomes (EXO-mCD24).**

A Exosome-based ELISA was used to measure expression of mCD24 in the purified exosomes. The expression level was compared to mCD24-negative exosomes and to mouse recombinant protein. The data are represented as average of three technical replicates ± SEM.

B Flow cytometry analysis of EXO-mCD24. EXO-mCD24 labeled with APC-conjugated anti-mCD24 antibody and gated to the exosome population.

C Western blot analysis of EXO-mCD24 exosomes. HSP70 was used as an exosomal marker.

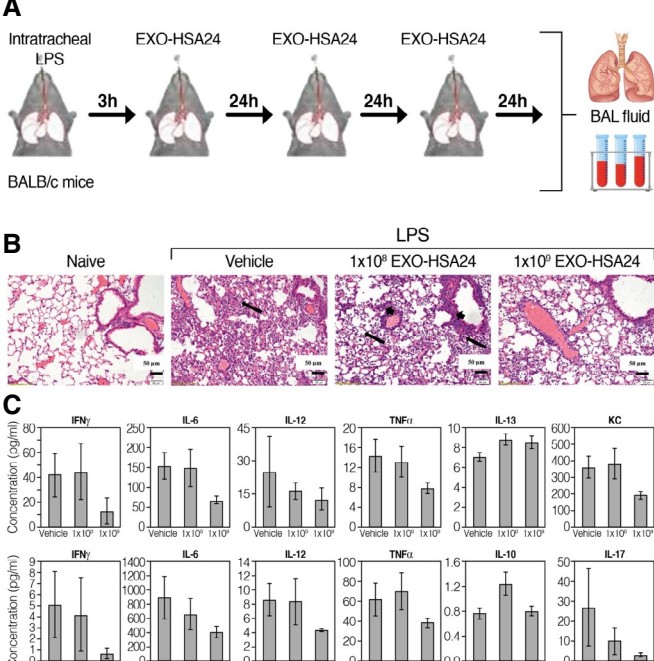

**Figure 5. EXO-mCD24 attenuates lung injury in a mice ARDS model.**

A Study design. Animals were challenged by intratracheal introduction of LPS, followed by EXO-HSA treatment, at $1 \times 10^8$ or $1 \times 10^9$ EXO-HSA/mouse, once daily for 3 days and then, mice were sacrificed. Serum and BAL fluid (Bronchial alveolar lavage fluid) were collected.

B Histopathology. Representative histological images of lung tissue. Tissues from saline and $1 \times 10^8$ EXO-HSA/mouse (low dose) treatment show extensive neutrophil infiltrate in the alveolar spaces (arrows) and around the bronchi and blood vessels (small arrows). In comparison, the inflammatory infiltrate in the $1 \times 10^9$ EXO-HSA/mouse (high dose) treatment is considerably attenuated. All images: hematoxylin and eosin (H&E) staining (naïve: no LPS challenge).

C The levels of systemic and local cytokines and chemokines. Serum (upper panel) and BAL fluid (lower panel) cytokines/chemokines levels. (KC/CXCL1, a neutrophil chemokine). Data are average ± SEM, $n = 10$ mice/group.

(SEM) mg/l. Thirty-two of the participants were white, one black, one Latin American, and one Asian.

## Safety

All adverse effects were classified as unrelated or probably not related to EXO-CD24 (Table 1). During the study period, 42 AEs irrespective of causality to the investigational product were reported for 29 (82.9%) patients. The AEs noted in ≥ 10% of the patients by MedDRA System Organ Class were investigations (11 events in 9 [25.71%] patients), followed in descending frequency order by blood and lymphatic system (6 events in 6 [17.14%] patients), nervous system (6 events in 5 [14.29%] patients), and general disorders (5 events in 5 [14.29%] patients). The only AE by MedDRA Preferred Term noted in ≥ 10% of the patients was leukocytosis (11.43%) related to the steroid therapy. The only reported SAE occurred in a patient enrolled in Group 1 (Patient 003). It was an acute asthmatic exacerbation in a patient with asthma. The event occurred toward the end of the follow-up period and improved dramatically, as expected, with the administration of ventolin and steroids inhalers. Another patient, who suffers from chronic lymphocytic leukemia and was immunocompromised due to rituximab therapy, completed the trial successfully with viral eradication but later was reinfected. One trial participant, in the $1 \times 10^9$ EXO-CD24/dose group, discontinued the treatment after 3 days due to sustained low oxygen saturation. The patient was transferred to intensive care for oxygen support and was released back to the ward after 4 days. After another 5 days, she was discharged to home care. The patient did not need ventilation support.

Seventeen of the 35 patients were PCR negative on day 7, and all of them negative at the end of the study (day 35). A follow-up of 369–501 days from the first patient to the last follow-up on February 17th did not disclose any drug-related adverse events. All subjects are alive and returned to routine activity although some of them are still suffering from post COVID-19 symptoms (weakness, shortness of breath on exertion, etc.). There were no deaths reported at the follow-up of 575 days.

## Efficacy

Efficacy can be evaluated only when a drug is compared with a placebo. However, between Day 0 and Day 5, according to the measurements performed in the morning prior to daily dosing, 57.1% (20/35) of the patients were able to reach a respiratory rate < 23 and 62.86% (22/35) had $SpO_2 > 94\%$ for two consecutive days. The mean (±SD) respiratory rate of the overall study population was 27.6 (±2.69) breaths/min at screening and it decreased to 15.6 (±0.5) on Day 5 ($P < 0.001$; Fig 7B). The mean (±SD) $SpO_2$ saturation levels of the overall population were 90.7 (±0.21%) at

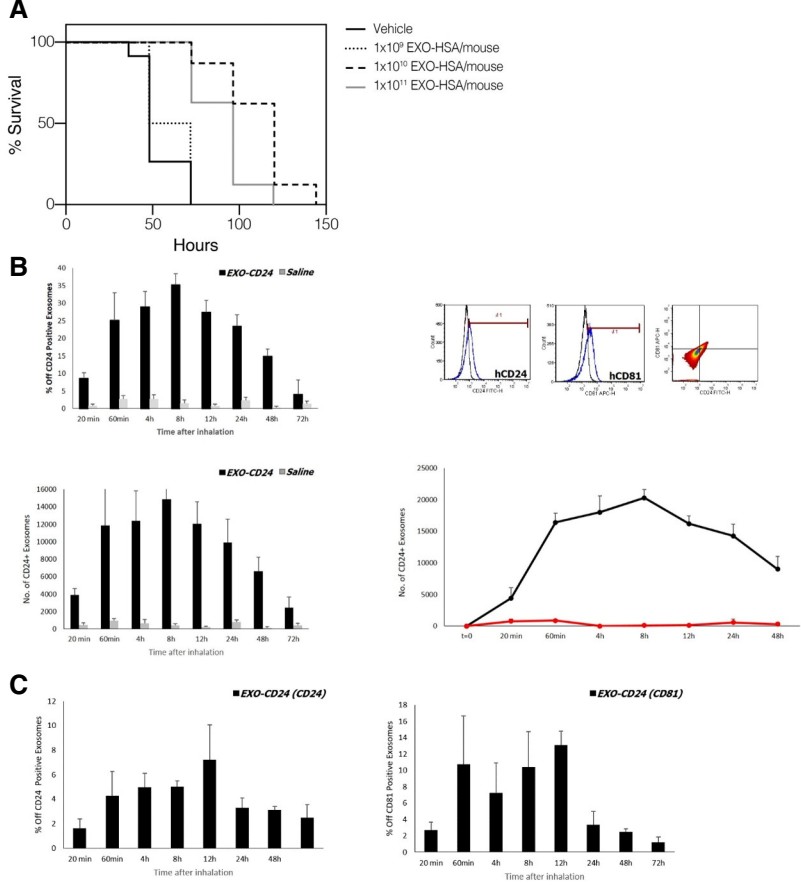

**Figure 6. *In vivo* PK of EXO-HSA and its efficacy in a mice sepsis model.**

A    Cecum ligation and puncture (CLP) was performed, followed by inhalation treatment with vehicle, or $1 \times 10^9$, $1 \times 10^{10}$, $1 \times 10^{11}$ EXO HSA/mouse, at 1, 8 and 24 h post-surgery.

B, C   EXO-CD24 was isolated from BALF (B) and serum (C) of treated and control mice. Evaluation of CD24/CD81-positive exosomes, at each time point, was carried out by flow cytometry. The data are presented as average ($n = 4$ mice at each time point) $\pm$ SEM.

screening, and increased to 96.1% ($\pm$01.3%; $P < 0.001$) on Day 5 (Fig 7C).

In addition, on Day 5 (prior to the fifth dose), the rate of $\geq 25\%$ increase in lymphocyte count was 70%, and the proportion of patients with a decrease of at least 20% in the NLR was 62.1%. These rates did not appear to be dose-dependent. Chest CT/X-rays on day 35 $\pm$ 2 demonstrated a reduction in lung abnormalities and opacity across all patients (representative image is shown in Fig 7D).

### COVID-19 global severity and symptoms questionnaire

At screening, all patients (35 of 35) had severe COVID-19 according to the global evaluation of severity. By Day 5, five patients (15.2%) still had severe COVID-19, while all others had mild (12.1%) or moderate (72.7%) disease. By Day 7, none of the patients had severe disease, while 21 (65.6%) had mild and 11 (34.4%) had moderate disease (Table 2).

On day 7, 80.7% (25 of 31) of the patients reported an improvement in the severity of their overall COVID-19-related symptoms at

their worst (Table 2). More than two-thirds of the patients displayed improvement in the severity of their shortness of breath (74.2%; 23 of 31), cough (67.7%; 21 of 31), feeling hot or feverish (67.7%; 21 of 31), and 58.1% (18 of 31) displayed an improvement in having low energy or tiredness.

### Systemic cytokine and CRP levels

There was a statistically significant decrease in C-reactive protein (CRP) level in the blood from a mean ($\pm$SEM) of 127.1 $\pm$ 14.7 mg/l on day 0 to 66.6 $\pm$ 10.4 mg/l on day 3 and 19.3 $\pm$ 7.7 mg/l ($\pm$SEM) ($P < 0.001$) on day 7 (Fig 8A). On day 35, CRP returned to normal level in all patients.

To evaluate the reduction in pro-inflammatory cytokines and chemokines following EXO-CD24 treatment in more detail, microarray studies were performed in patients' plasma at different time points. MIF 3a, IL-17A, IL-1β, IL-6, TGFα, TNFα, and IL-1α levels were significantly reduced on days 4, 7 and 35 in comparison with pretreatment levels (day 0). Reduction of cytokine levels in eight patients is shown in Fig 8B. For the rest of the patients, an

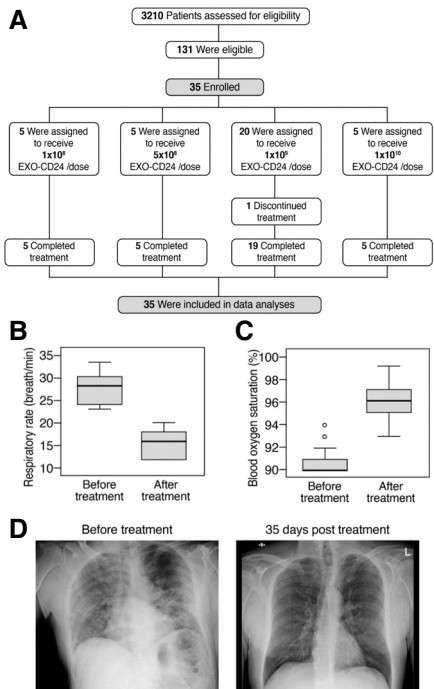

**Figure 7. Enrollment and clinical results.**

A   One trial participant, in the $1 \times 10^9$ EXO-CD24/dose group discontinued the treatment after 3 days due to sustained low blood oxygen saturation.

B, C   (B) respiratory rate and (C) Blood oxygen (SpO$_2$) and before treatment (Day 1, measured before administration of first dose) and after treatment (Day 7). Descriptive statistics (minimum, maximum, median, first quartile, third quartile) were calculated for Blood oxygen (SpO$_2$) and respiratory rate. These are presented graphically via box plots before and after treatment.

D   Representative X-ray image.

additional panel of analytes (Quantibody Multiplex ELISA Array, RayBiotech, Almog Diagnostic Ltd, Israel) demonstrated a significant reduction in a time-dependent manner (Fig 8C).

In some patients, IL-10 level increased, while in others, it decreased. The dramatic early rise in IL-10, canonically classified as an anti-inflammatory cytokine, appears to be a distinguishing feature of hyperinflammation during severe SARS-CoV-2 infection (Lu *et al*, 2021) and several studies indicate that IL-10 levels predict poor outcomes in patients with COVID-19.

IL-6 monitoring, using the BD cytometric bead array, from day 0 to day 35, showed a significant decrease among all the patients tested (Fig 8D).

## Discussion

EXO-CD24 is a new precision nanotechnology aimed to target the most important complication of COVID-19, the cytokine storm in the lungs. EXO-CD24 consists of two breakthrough technologies. CD24 is the drug and exosomes are the carriers. EXO-CD24 is given by inhalation directly into the lungs, to suppress the body's inflammatory response to tissue injuries. The potential efficacy

stems from using an endogenous immunomodulator of the immune system (CD24). Safety has been secured by using natural exosomes, a promising endogenous drug carrier. Exosomes are not expected to rouse side effects, and they are being used in tens of on ongoing clinical trials.

CD24 binds DAMPs and inhibits the activation of the NFκB pathway (the main pathway for cytokines and chemokines production) in two critical points (TLR and Siglec-10). It does not bind PAMPs; hence, it does not interfere with pathogen clearance. In contrast to anti-inflammatory therapies that are cytokine-specific (anti-IL-6) or steroids that shut down the entire immune system, the activity of CD24 is upstream which affects arrays of cytokines reverting them back to normal activity. Exosomes serve as a drug delivery system. They are intraluminal nano-vesicles, naturally present in body fluids, that play an important role in intercellular communications. Through inhalation, EXO-CD24 enhances its effectiveness by directly reaching the lungs. EXO-CD24 can be produced effectively, rapidly, at low cost, and tech transfer is easy.

CD24 is of particular interest as a therapeutic agent for virus-induced hyper-inflammation and ARDS. It reverses abnormal and uncontrolled activity of the NF-kB pathway, without interfering with viral clearance activity.

Preclinical studies support the safety and therapeutic potential of CD24 by exosomal delivery. Toxicology evaluation in mice revealed no adverse effects or negative signs of EXO-mCD24 on animal weight, hematology, chemistry, urine tests, and organ histology, describing potentially positioning EXO-mCD24 as a safety drug at both doses tested.

The key point of this phase Ib/IIa study is safety. A typical phase I study includes, in addition to safety data, pharmacokinetic data and measurements of drug levels and distribution of the drug in target organs. Unfortunately, it is impossible to track an inhaled drug. Moreover, as EXO-CD24 is composed of human exosomes and CD24 that cannot be distinguished from the endogenous ones, its level cannot be measured systemically. The decrease in serum cytokine arrays and inflammatory markers, in patients that participated in the phase Ib/IIa study, in a time-dependent manner, suggest the efficacy of EXO-CD24. In addition, CD24 pharmacokinetics has been assessed in mice. After a single dose of inhaled exosomes, EXO-CD24 exosomes were able to be detected in a typical pattern in the BALF and even in the bloodstream, without accumulation in other organs. Efficacy of EXO-mCD24 stems from experiments conducted in several large-scale animal models; (1) In LPS-induced ARDS mouse models, (2) cecum ligation and puncture induced sepsis model. EXO-mCD24 inhalation ameliorated, in a dose-dependent manner, lung injury, reduced local and systemically cytokine/chemokine release (in the BALF and serum, respectively). We identified that, after 72 h, administration of high doses of EXO-mCD24 ($1 \times 10^9$ exosomes/mice) was more effective in reducing pulmonary inflammation than lower doses ($1 \times 10^8$ exosomes/mice).

The study of COVID-19 disease in rodents has been hindered by the incompatibility of ACE2 with virus spike proteins in different species. There have been previous reports of LPS, or endotoxins, being involved in ARDS pathophysiology. The resulted cytokine storm, pneumonia, and coagulopathy are commensurate with moderate and severe COVID-19 disease noted in humans. There have been no studies reported to date showing that animal models can reproduce the severity of COVID-19 disease when infected with wild

Table 1.  Summary of adverse events according to EXO-CD24 treatment group.

| Event or abnormality | 1 × 10⁸ EXO-CD24 Group (N = 5) | 5 × 10⁸ EXO-CD24 Group (N = 5) | 1 × 10⁹ EXO-CD24 Group (N = 5) | 1 × 10¹⁰ EXO-CD24 Group (N = 5) |
|---|---|---|---|---|
| Any adverse event- no. of patients (%) | 4 | 5 | – | 14 (5) |
| Abnormal liver function tests n (%) | – | 1 (20) | 1 (5) | 1 (20) |
| Asthma exacerbation n (%) | 1 (20) | – | – | – |
| Cellulitis n (%) | – | 1 (20) | – | – |
| Cough on exertion n (%) | – | – | 1 (5) | – |
| Dizziness n (%) | – | – | – | 1 (20) |
| Dyslipidemia n (%) | – | – | – | 1 (20) |
| Fever n (%) | – | – | – | 1 (20) |
| Hair loss n (%) | – | – | 1 (5) | – |
| Headache n (%) | – | – | 4 (20) | – |
| Increased appetite n (%) | – | – | – | 1 (20) |
| Leukocytosis n (%) | – | 1 (20) | 3 (15) | – |
| Loss of appetite | – | – | – | – |
| Loss of sense of taste | 1 (20) | – | – | – |
| Lower back pain | – | – | 1 (5) | – |
| Mild lymphopenia | – | – | 3 (15) | – |
| Nausea | – | – | – | 1 (20) |
| Pain in site of phlebotomy | – | – | 1 (5) | – |
| Palpitations | 1 (20) | – | – | – |
| Panic attack | – | – | 1 (5) | – |
| Respiratory failure | – | – | 1 (5) | – |
| Shortness of breath on exertion | – | – | 1 (5) | – |
| Suspected hyperkalemia | – | – | – | 1 (20) |
| Thrombocytosis | 1 (20) | – | – | – |
| Weakness | – | 2 (40) | – | 1 (20) |

Table 2.  COVID-19 global severity, by dose group and overall.

| | | 1 × 10⁸ (N = 5) | | 5 × 10⁸ (N = 5) | | 1 × 10⁹ (N = 20) | | 1 × 10¹⁰ (N = 5) | | Total (N = 35) | |
|---|---|---|---|---|---|---|---|---|---|---|---|
| | | N | % | N | % | N | % | N | % | N | % |
| | Global Evaluation of severity | 5 | 100.00 | 5 | 100.00 | 20 | 100.00 | 5 | 100.00 | 35 | 100.00 |
| Screening | 4. Severe | | | | | | | | | | |
| Day 5 | 2. Mild | – | – | – | – | 2 | 11.011 | 2 | 40.00 | 4 | 12.12 |
| | 3. Moderate | 5 | 100.00 | 3 | 60.00 | 14 | 77.78 | 2 | 40.00 | 24 | 72.73 |
| | 4. Severe | – | – | 2 | 40.00 | 2 | 11.11 | 1 | 20.00 | 5 | 15.15 |
| Day 7 – Follow-up | 2. Mild | 3 | 60.00 | 4 | 80.00 | 10 | 55.56 | 4 | 100.00 | 21 | 65.63 |
| | 3. Moderate | 2 | 40.00 | 1 | 20.00 | 8 | 44.44 | – | – | 11 | 34.38 |

SARS-CoV-2 (Rittirsch *et al*, 2008; Wang *et al*, 2008; Al-Ani *et al*, 2022).

Furthermore, we found that administration of EXO-mCD24-displayed exosomes significantly increased survival in a CLP-animal model of sepsis.

Efficacy must still to be confirmed versus placebo; however, it was exciting and promising to see the potential efficacy of the drug.

(1) a subjective improvement in clinical parameters (respiratory rate, oxygen saturation, and early home discharge) as experienced by the medical staff. (2) a subjective improvement described by the patients including a self-reported questionnaire. (3) an improvement in chest X-ray/CT. (4) an impressive improvement in all inflammatory blood indices. (5) arrays of cytokines and chemokines in patients' blood reverting back to normal levels.

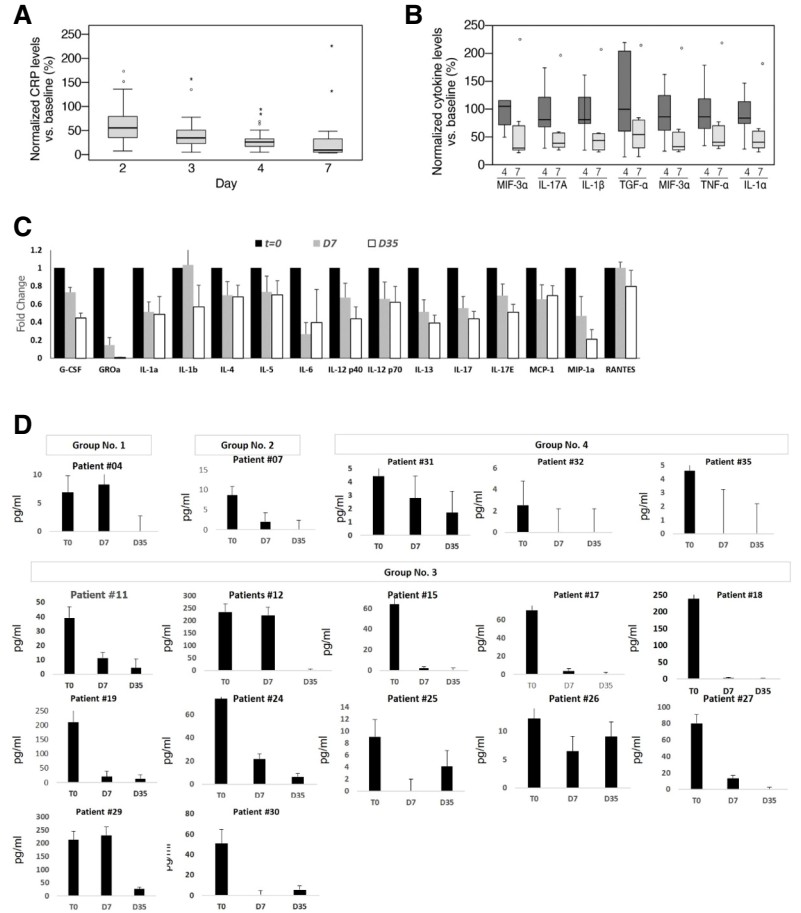

**Figure 8. Patients' systemic inflammatory indices.**

A  Systemic CRP levels. Values are normalized against baseline value (Day 0). Descriptive statistics (minimum, maximum, median, first quartile, and third quartile) were calculated for CRP levels. These are presented graphically via box plots before and after treatment (*n* = 35 patients).

B  Systemic cytokine levels. Cytokine levels were measured during (Day 3/4) or after (Day 7/8) treatment with EXO-CD24 using Quantibody Multiplex ELISA Array. Values are normalized against baseline value (Day 0). Samples from 11 patients were analyzed. Results from eight patients are shown (results for three patients are not shown because one or more baseline results were below detection level). Box plot of Systemic cytokine levels by days after treatment (minimum, maximum, median, first quartile, third quartile).

C  During the course of EXO-CD24 treatment and follow-up, cytokine levels were measured for the rest of the patients (*n* = 24 patients). Values represent the fold change from day 0, mean ± SEM.

D  Monitoring systemic IL-6 cytokine level following EXO-CD24 treatment. Samples from 17 patients, representing the four experimental groups, were analyzed using the BD™ Cytometric Bead Array (CBA). Values represent mean ± SD.

Cytokine storm is the main cause of morbidity and mortality following other viral infections, for example, SARS-CoV-1, MERS (Soy *et al*, 2021), and influenza (Tang *et al*, 2020). Preclinical studies support the therapeutic potential of CD24 by exosomal delivery. Tian *et al* (2018) have previously studied CD24Fc, a recombinant fusion protein composed of the extracellular domain of CD24 linked to a human immunoglobulin G1 (IgG1) Fc domain. They found that CD24Fc reduces lung pathology and the incidence of viral induced pneumonia in an animal model (Tian *et al*, 2018), herein supporting our conclusion.

As of April 27, 2022, there are 510 million COVID-19 cases worldwide, with more than 6 million deaths. The prognosis in 95% of patients is self-resolution without treatment or with only supportive care. Overreaction of the immune system (cytokine storm) is a major complication that, in 5% of COVID-19 patients, leads to

pulmonary insufficiency, the need for ventilation and even death. Other therapeutic regimens on the market can be used to combat the COVID-19 pandemic; however, they come with significant drawbacks. Vaccinations lose their efficacy after a few months, and many do not have access to them. The emergence of new mutations has sparked public concern regarding further booster shots and their utility. Home therapy is obviously attractive; however, antiviral drugs have many associated obstacles to overcome, they are expensive, there is a short window for intervention and many patients will receive a therapy that they do not need. In addition, a requirement for Pfizer's antiviral pill is that it should be taken with ritonavir, a drug used to treat HIV. This would make the treatment unsuitable for many people with preexisting medical conditions. The issue with antivirals, like those mentioned above, is that they would have to be used before people are considered ill enough to need therapy

other than symptomatic self-care treatment. The proportional reduction in the risk of hospitalization or death is impressive. However, the absolute risk reduction from 14 to 7% means that quite a lot of people need to be treated to prevent one hospitalization or death.

COVID-19 remains a life-threatening disease especially to patients that are immunocompromised. Moreover, inflammatory lung disease is an array of many lung diseases (such as COVID-19, severe influenza, ARDS, and more) affecting more than a billion patients worldwide with millions of associated deaths. No effective treatment is currently available. It is apparent that a drug for the cytokine storm, COVID-19, and other diseases with a hyperactive immune system is urgently needed.

Several preclinical studies for the treatment of other inflammatory lung diseases (such as pulmonary fibrosis, asthma, and sepsis) were carried out. Preliminary results strengthen our hypotheses about the effectiveness of EXO-CD24 in treating other lung diseases. The EXO-CD24 platform can be used for the treatment of additional respiratory and systemic diseases associated with over-activation of the immune system, with a high mortality rate and similar pathologies without an appropriate therapy to date.

This phase Ib/IIa clinical trial of EXO-CD24 is reassuring in confirming the safety and potentially promising efficacy of the 5-day treatment regimen, with no drug-related adverse effects observed regardless of the dose group. These results gave the necessary support to move on to a phase IIb, dose finding study in three medical centers in Athens. However, we caution against over interpretation from single-arm studies.

This study suggests that EXO-CD24 is tolerable even at high doses of $10^{10}$ EXO-CD24. The potential efficacy justifies further clinical investigations of this substance in COVID-19 patients, as well as, for other indications with significant unmet therapy.

# Materials and Methods

### Cell lines

Human monocyte cell line U937 was cultured at 37°C in a humidified atmosphere of 95% air and 5% $CO_2$ in RPMI-1640 medium (Biological Industries, Beit HaEmek, Israel) supplemented with 10% fetal bovine serum (FBS), 1% penicillin, and 1% streptomycin (Biological Industries).

### mCD24 plasmid construction

A DNA fragment coding for a murine CD24 (HSA) fragment was amplified by PCR using the plasmid pHR'CMV-HSA as a template using primers NheI-kozak-HSA F- (5'-ATATATGCTAGCGCTACCGG ACTCAGATCTgCC atgggcagagcgatgg-3') and HSA-EcoRI R- (5'-TAT ATGAATTCGAAGCTTGAGCTC gtactaacagtaga gatgtagaag-3'). The PCR product was digested by NheI and EcoRI and inserted into the pIRES-GFP plasmid, which was cleaved with the same enzymes. The resulting plasmid was named CD24/HSA-IRES-GFP.

### EXO-mCD24

Transient transfection of Expi293™ cells (Thermo Fisher Scientific) in suspension culture was used to prepare homologous EXO-mCD24

for all the other nonclinical experiments. The process is described below.

Expi293 cells were cultured in a 30 ml Expi293 expression medium (Gibco, A14351-01) to a density of $0.5 \times 10^6$/ml in Erlenmeyer flask. Cell growth is maintained in 37°C humidified incubator, 8% $CO_2$ on orbital shaker at speed of 110 RPM. Transfection is performed according to the transfection kit manufacturer's protocol (ExpiFectamine™ 293 Transfection Kit, A14524). Medium is harvested 5 days following transfection, then centrifuged at 200 $g$ for 5 min to remove cells and the supernatant is filtered through a 0.2um filter to remove cell debris. Exosomes were purified from culture media as described in section "Preparation of CD24 displaying exosomes."

### Preparation of the CD24 displaying exosomes

The active pharmaceutical ingredient under study was exosomes engineered to overexpress CD24 protein. As previously described (Shapira et al, 2011), the human *CD24* gene was cloned downstream to two tetracycline-operator sequences, resulting in pCDNA4/TO-CD24, which was transfected into the human embryonic kidney T-REx-293™ cells (Thermo-Fisher Ltd., USA).

To isolate CD24-displaying exosomes, a total amount of $7 \times 10^7$ cells were seeded in a cell factory system (50% confluence), in a total volume of 200 ml complete medium supplemented with 1 µg/ml tetracycline. 5% of USDA-approved serum was added. After 48 h of incubation, the growth medium was removed and cells were resuspended in phosphate buffered saline (PBS). After washing, 200 ml of serum- and protein-free Expi293™ medium (ThermoFisher Scientific) supplemented with 1 µg/ml tetracycline was added and cells incubated for 72 h (37°C, 5% $CO_2$). Following incubation, the growth medium was collected into 50-ml tubes and centrifuged at 3,000 × $g$ for 15 min (4°C) to remove cells and cell debris. The supernatant was filtered using a 0.22 µ pore size filter. ExoQuick-CG (SBI, System Biosciences) exosome precipitation solution was added (3.3 ml/10 ml), mixed by gentle inversion, and the tubes were refrigerated overnight (at least 12 h). The following day, the ExoQuick-CG/biofluid mixture was centrifuged at 1,500 × $g$ for 30 min, at 4°C, and the supernatant was aspirated. Residual ExoQuick-CG solution was removed by centrifugation at 1,500 × $g$ for 5 min, followed by aspiration of all traces of fluid. The pellet containing the exosomes was resuspended in saline and transferred to a dialysis cassette. Dialysis was performed against 5 l of freshly prepared PBS, overnight, 4°C. The exosomes were transferred into an Amicon tube (10,000 MW) and centrifuged at 15°C until preferred volume was reached. The concentration of purified exosomes was determined, and the preparation was diluted as needed, filtered (using a sterile 0.22 µ pore size filter), and stored at −80°C.

### Exosomes concentration, nanoparticle tracking analysis

The concentration and particle size of the purified exosomes are measured by a nanoparticle tracking analysis (NTA) method using a NanoSight NS300 device that characterizes nanoparticles in solution. Particles suspended in a liquid are loaded into the laser module sample chamber and viewed in close proximity to the optical element. The NTA software tracked several particles

individually and uses the Stokes-Einstein equation to calculate the hydrodynamic diameter of the particles. Each particle is individually but simultaneously analyzed by direct observation and measurement of diffusion events. The laser module contains thermoelectric Peltier elements, allowing the sample temperature to be controlled.

Before NTA measuring, an aliquot of the isolated vesicles is thawed at room temperature and diluted 100 times in saline. The test sample is then filled in a 1 ml disposable syringe, and after removing air bubbles, the syringe is directly connected to a luer port on the top-plate. The test chamber is filled slowly against gravity, allowing any bubbles to escape. The measurements were performed at least twice.

### Flow cytometry analysis of CD24 exosomes

Purified exosomes were incubated with 10 μg/ml NS17-FITC (produced in house), anti-CD81-APC (Almog Diagnostic, Shoham, Israel) or were left unstained for 60 min on ice protected from light. Mix of stained beads in different sizes (100, 200, and 500 nm) were used as an internal control to verify the size of the nanoparticles. Analysis was performed using CytoFlex Benchtop Flow Cytometer (Beckman Coulter Life Sciences).

### *In vitro* PMA stimulation

U937 cells were seeded in 24-well plates at $80 \times 10^3$ cells/well followed by overnight incubation at 37°C, 5% $CO_2$. Cells were stimulated with Phorbol 12-myristate 13-acetate (PMA; Sigma) at a concentration of 100 ng/ml for 72 h. 10 μg/ml hrHMGB1 and EXO-CD24 were then added, and after 24 h, the medium was collected and the cytokine levels were analyzed using a Luminex's xMAP Cytokine "Multi-plex" array (Cat. Num LKTM014, Human XL Cytokine Discovery Fixed Panel).

### Animal husbandry

Mouse studies were performed by Science in Action (SIA; Ness Ziona, Israel), a contract research organization (CRO). All experimental protocols were approved by the Israel Board for Animal Experiments and in compliance with the Israel Animal Welfare Act as well as the Institutional Ethics Committee. Animals were purchased from Envigo (Jerusalem, Israel); number of animals were as detailed per experimental model below. Animal handling was performed according to the guidelines of the National Institute of Health (NIH) and the Association for Assessment and Accreditation of Laboratory Animal Care (AAALAC).

### Experimental design—dose toxicity assay

EXO-CD24 was administered to 8-week-old female Balb/c mice ($n = 30$), daily, for 5 days, by inhalation of 200 μl aerosolized exosomes using an inhalation cage. Saline was used as control treatment. Each treatment was divided into two groups. In the first one, mice were killed 24 h post final-dose and in the second, mice were followed up for an additional week (follow-up mice). Detailed clinical observation was carried out prior to dosing, every 30 min for the first 3 h after the first dosing, daily thereafter up to finishing, and

twice weekly for the follow-up mice group. Animals were weighed daily up to termination. Follow-up animals were additionally weighed three times a week during the follow-up period. Ophthalmoscopic examination was carried out twice at the pretest and before termination. Urine and blood were collected on the final day and analyzed for hematology and blood chemistry. On the day of termination, the following organs were collected: brain, heart (sections of left and right ventricles and atria, septum with papillary muscle), kidneys, liver, lungs, spleen, thymus, and thyroid. All organs were weighed and tissues from the vehicle and the EXO-CD24/mouse groups were processed to slides and evaluated by a certified pathologist by microscopy analysis (Patho-Logica Ltd, Ness Ziona, Israel).

### LPS-induced ARDS mice model

ARDS was induced in 8-week-old female BALB/C mice ($n = 35$). Briefly, mice were anesthetized, orally intubated with a sterile plastic catheter and challenged with intratracheal instillation of 800 μg of lipopolysaccharide (LPS; *E. coli* origin, serotype 055: B5; ChemCruz, Batch/lot No.: C3120) dissolved in 50 μl of PBS. Naive mice (without LPS instillation) served as controls. EXO-mCD24/HSA treatment consisted of daily inhalation of aerosolized exosomes via the endotracheal tube with initial dose given at 3 h after LPS administration. The study was finished 72 h after LPS challenge. On the final day, serum bleeding was performed and bronchial alveolar lavage (BAL) fluid was collected. BAL fluid and serum were analyzed for cytokines/chemokines levels. Lung tissue was preserved in 4% formaldehyde for histopathology analysis.

### Cecum ligation and puncture (CLP)-induced sepsis model

Eight-week-old male C57bl mice ($n = 53$) were anesthetized with Ketamine/Xylazine (Sigma-Aldrich, Rehovot, Israel). A 1 cm ligation of the cecum, one puncture using 21-gauge needle, extruding of 1 cm of fecal matter was performed. All the animals were sc injected with 1 ml of saline, immediately following surgery, and then every 12 h for 3 days. Mice were then treated with EXO-m24 aerosolized exosomes, administered for 20 min in an inhalation cage, at 1-, 8- and 24-h post-surgery. Animals were weighed daily. Thirty-six hours after surgery, four mice of from each group were sacrificed to collect blood, lung fluid (samples were stored in at −80°C until analyzed), and lung tissue (preserved in 4% formaldehyde). Remaining animals in each group ($n = 8$) were followed up for survival. Survival across treatment groups was estimated by the Kaplan–Meier method, and compared with the log-rank test. Cox regression was used to estimate relative hazards and 95% confidence intervals for mortality over time across treatment groups. The proportional hazards assumption was tested, and no violations were detected. SPSS 27 (IBM, New York, USA) was used for all data analyses in this study.

### PK/PD study

PK/PD were studied in 8-week-old female BALB/C mice ($n = 132$). Briefly, mice received one inhaled dose ($10^9$, $10^{10}$ or $10^{11}$) of aerosolized exosomes, in 20 min, via inhalation cage.

At the end of the inhalation (20 min) four mice from each group were sacrificed at 20, 60 min, 4, 8, 12, 24, 48, and 72 h. At each time point, blood and lung fluid were collected and stored in 80°C and organs (heart, kidney, brain, spleen, liver, and kidneys) were immediately frozen in liquid nitrogen. The collected organs were transferred to extraction with 0.7 ml of saline in a bullet system and the extraction liquid was separated and kept in an 80°C.

## Statistical analysis

Descriptive statistics (N, mean, median, standard deviation, and proportion) were calculated for study variables, and these are presented in tables as well as graphically via box plots. Independent-samples $t$-test for continuous variables and chi-square tests for categorical variables was used to compare between intervention and control groups. All tests were two-tailed and $P < 0.05$ were considered significant. Statistical analysis was performed using GraphPad Prism. Data are presented as mean ± SEM, unless individual measurements are shown; $P < 0.05$ was considered to be statistically significant.

# Patients

## EXO-CD24: test product, dose, mode of administration, and batch number

EXO-CD24 investigational drug preparation was carried out as described above.

EXO-CD24 was produced in a clean room, c-GMP-compliant manufacturing facility at Accellta Ltd (Haifa, Israel). It was diluted in normal saline at doses of $1 \times 10^8$, $5 \times 10^8$, $1 \times 10^9$ and $1 \times 10^{10}$ particles and provided in 2 ml amber glass vials. Prior to administration, 1.5 ml saline (0.9% sodium chloride Intravenous Infusion BP) are added, and the total volume of suspension is transferred to the nebulizer chamber, standard hospital-grade inhalation device, for inhalation. The initial dose ($1 \times 10^8$ particles) was based on similar studies using viruses as the therapeutic vehicle, as well as on a study aimed at evaluating the safety and tolerance of inhaled mesenchymal-cell-derived exosomes in healthy volunteers (NCT04313647). In addition, the doses were found to be effective in preclinical studies.

Analysis and examination of EXO-CD24 secreting cells (Trex293-CD24) and the resulting EXO-CD24 investigational drug substrate and drug product were characterized and tested, for safety and efficacy, using many different analytical and biological tests including identity, purity, content, potency, microbial (mycoplasma, sterility, and endotoxin LAL), and viral contamination validation tests.

EXO-CD24 (batch# 03) was used in the clinical trial.

## Clinical trial oversight

This is an investigator initiated, open-label, single-arm, phase Ib/IIa clinical study, of patients with moderate/high severity COVID-19 infection treated with EXO-CD24. The trial was monitored by Chana Debbie Sternberg Ltd, a clinical trial monitoring and auditing company.

## Diagnosis and main criteria for inclusion/exclusion

Included patients were male and female, with moderate/high severity of COVID-19 disease.

1. A COVID-19 diagnosis confirmed with a SARS-CoV-2 viral infection-positive polymerase chain reaction (PCR) test
2. Age 18–85 years
3. Disease severity: moderate/severe according to the following criteria (at least one clinical parameter and one laboratory parameter were required):

   a. Clinical evaluation

       i. Respiratory rate ≥ 23/min and ≤ 30/min
      ii. SpO$_2$ at room air ≤ 94% and ≥ 90%
     iii. Bilateral pulmonary infiltrates > 25% within 24–48 h or a severe deterioration compared with imaging at admission

   b. Evidence of an exacerbated inflammatory process

       i. LDH ≥ 450 u/l
      ii. CRP ≥ 100 mg/l
     iii. Ferritin ≥ 1,650 ng/ml
      iv. Lymphopenia ≤ 800 cells/mm$^3$
       v. D-dimers > 1
      vi. Fibrinogen ≥ 450 mg/l

4. Willing and able to sign an informed consent

   Exclusion criteria included:

   (1) Age < 18 years or > 85 years; (2) Any concomitant illness that, based on the judgment of the investigator, is terminal; (3) Any concomitant illness that, based on the judgment of the investigator might affect the interpretation of the study results (i.e., immune-deficient patients); (4) Mechanically ventilated patient or those who will probably require ICU admission or mechanical ventilation within 24 h from enrollment; (5) Pregnancy (positive urine pregnancy test [women of childbearing potential only]) or breast feeding; (6) Unwilling or unable to provide informed consent; (7) Active cancer and (8) Participation in any other study in the last 30 days.

## Safety

Safety was reviewed weekly by an independent protocol safety review team. Trial details, including inclusion and exclusion criteria, were deposited at https://www.clinicaltrials.gov/ct2/show/NCT04747574 and can be accessed anytime.

EXO-CD24 was given as an add-on treatment to standard of care (Table 3). EXO-CD24 was stored at −80°C, and thawed on ice prior to treatment, for at most 2 h. The therapeutic was diluted, at the doses indicated below, in saline solution for inhalation and given once daily (QD) for 5 consecutive days.

## Safety end points

Safety end points included monitoring of asked local and systemic adverse events (AE) during conflPrimary safety endpoints included the following: (1) incidence of treatment (dose)-related serious

**Table 3. Clinical characteristics of the patients as baseline.**

| EXO-CD24 Dose | | | | | |
| --- | --- | --- | --- | --- | --- |
| Treatment group | $1 \times 10^8$ | $5 \times 10^8$ | $1 \times 10^9$ | $1 \times 10^{10}$ | Total |
| Dexamethasone | 4 | 5 | 15 | 5 | 29 |
| Remdesivir | 4 | 3 | 9 | 2 | 18 |
| Clexane | 5 | 5 | 18 | 5 | 33 |
| Mean blood oxygen saturation (range) – % $SpO_2$ | 91.2 (90–94) | 90.4 (90–92) | 90.7 (90–93) | 90.0 (90–92) | 90.7 (90–94) |
| Mean respiratory rate (range) – breaths/min | 28.8 (28–30) | 29.4 (28–30) | 27.4 (23–30) | 25.0 (23–27) | 27.6 (23–30) |
| Mean Blood CRP concentration (SEM) – mg/l | 88.4 ± 41.9 | 143.6 ± 8.4 | 139.4 ± 23.0 | 108.8 ± 21.3 | 127.1 ± 14.7 |

adverse events; (2) incidence of all adverse events related or unrelated to the study treatment.

The primary safety parameters were (1) bronchospasms, (2) superinfection, (3) severe clinical deterioration, (4) ventilation, (5) all-cause mortality, and (6) and viral load. Safety data are included up to the cutoff date of March 14th, 2021. The patients are continuing to be followed up monthly. Data are available up to 575 days (April 27th, 2022).

### Efficacy end points

Exploratory efficacy end points of EXO-CD24 included the following: (1) reducing respiratory rate; (2) increasing blood oxygen saturation ($SpO_2$); and (3) reducing the level of pro-inflammatory cytokines.

Exploratory end points included the following: (1) alive at day 7 without bronchospasms, unexpected infections, or a significant clinical deterioration compared to baseline; (2) proportion of patients with respiratory rate < 23/min for 24 h; (3) decrease/improvement in respiratory rate from baseline to Day 7; (4) proportion of patients with $SpO_2$ saturation > 94% for at least 24 h; (5) increase/improvement in $SpO_2$ saturation from baseline to Day 7; (6) proportion of patients with no artificial ventilation after 7 days of treatment; (7) proportion of patients with a decrease by 50% in either CRP/LDH/fibrinogen/ferritin/IL-6, D-dimers from baseline to Day 7 (8) proportion of patients with an increase of 25% in the absolute lymphocyte count, sustained for ≥ 48 h to Day 7; (9) change in the absolute lymphocyte count from baseline to Day 7; (10) proportion of patients with a decrease of 20% in the NLR, sustained for ≥ 48 h to Day 7; (11) change in the LNR from Baseline to Day 7; (12) rate of categorical and absolute score improvement of Covid-19 status on Day 7 improving from "Severe" to at least "Moderate" or from "Moderate" to "Moderate–Mild" using any of several COVID-19 clinical severity ordinal scales: NIAID-OS (8-point), WHO-OS (7-point), WHO-OS (10-point) in each dose group and the total population; (13) death rate at end of study (Day 35); (14) death rate at end of follow-up (up to day 575); proportion of patients with no mechanical ventilation (ECMO, NIV, high flow) on Day 7; (15) proportion of patients with hemodynamic instability or requiring vasopressors on Day 7; (16) proportion of patients requiring admission to an Intensive-Care Unit (ICU) by Day 7. Efficacy data are included up to the cutoff date of March 14th, 2021, and all-cause mortality by day 575 (April 27th, 2022) of follow-up.

### Clinical and laboratory analysis

Vital signs were taken at each study visit (Day 1–5, 7, and 35), including body temperature, pulse, respiratory rate, SpO2, and blood pressure. Chemistry, hematological, and biochemical analyses were performed on Day 1, 3, 5, 7, and 35. A chest CT/X-ray was taken as part of the screening activities, unless unavailable within 7 days of screening.

Cytokine levels were analyzed using the Human XL Cytokine Discovery Luminex® (R&D Systems) and the BD™ Cytometric Bead Array (CBA).

### Human XL cytokine discovery Luminex®

Several-defined target analytes were chosen and simultaneously profiled using the magnetic Luminex High-Performance Assay, a flexible bead-based multiplex assay. Briefly, analyte-specific antibodies were precoated onto color-coded microparticles. Microparticles, standards, and samples were pipetted into wells and the immobilized antibodies bound the analytes of interest. After washing away any unbound substances, a biotinylated antibody cocktail specific to the analytes of interest was added to each well. Following a wash to remove any unbound biotinylated antibody, Cy3 equivalent dye-conjugated streptavidin, which binds to the biotinylated detection antibodies, was added to each well. A final wash removed unbound streptavidin-Cy3 and the microparticles were resuspended in buffer and read using the Luminex analyzer. Then, the RayBio Array Analysis Tool, an-array-specific excel-based program, performed sophisticated data analysis of the raw numerical data extracted from the array scan.

### BD™ Cytometric bead Array

The BD™ Cytometric Bead Array, a flow cytometry application, was used to measure the levels of IL-6 in patients' plasma. Following the preparation and dilution of the standards and mixing of the capture beads, these reagents and test samples were transferred into the appropriate assay tubes for incubation and analysis.

### Trial oversight and IRB approval

The study, the protocol, and consent forms were approved by the Tel Aviv Medical Center Ethics Committee and the Israel Ministry of Health (IMOH), in accordance with GCP guidelines. In this study, the inclusion of a placebo group was not approved by the Israeli Ministry of Health. All participants provided written informed

**Table 4. Changes in COVID-19 symptoms. Questionnaire on Day 7, by dose group and overall.**

| | Day 7 – Follow-up | | | | | | | | | |
|---|---|---|---|---|---|---|---|---|---|---|
| | $1 \times 10^8$ (N = 5) | | $5 \times 10^8$ (N = 5) | | $1 \times 10^9$ (N = 20) | | $1 \times 10^{10}$ (N = 5) | | Total (N = 35) | |
| | N | % | N | % | N | % | N | % | N | % |
| **A** | | | | | | | | | | |
| **Stuffy or runny nose** | | | | | | | | | | |
| Worsened | – | – | 1 | 25.00 | 1 | 5.56 | – | – | 2 | 6.45 |
| No change | 5 | 100.00 | 3 | 75.00 | 12 | 66.67 | 2 | 50.00 | 22 | 70.97 |
| Improved | – | – | – | – | 5 | 27.78 | 2 | 50.00 | 7 | 22.58 |
| **Sore throat** | | | | | | | | | | |
| Worsened | – | – | – | – | 2 | 11.11 | – | – | 2 | 6.45 |
| No change | 5 | 100.00 | 4 | 100.00 | 14 | 77.78 | 3 | 75.00 | 26 | 83.87 |
| Improved | – | – | – | – | 2 | 11.11 | 1 | 25.00 | 3 | 9.68 |
| **Shortness of breath (difficulty breathing)** | | | | | | | | | | |
| Worsened | – | – | – | – | – | – | – | – | – | – |
| No change | 1 | 20.00 | 1 | 25.00 | 5 | 27.78 | 1 | 25.00 | 8 | 25.81 |
| Improved | 4 | 80.00 | 3 | 75.00 | 13 | 72.22 | 3 | 75.00 | 23 | 74.19 |
| **Cough** | | | | | | | | | | |
| Worsened | – | – | – | – | – | – | – | – | – | – |
| No change | 2 | 40.00 | 2 | 50.00 | 4 | 22.22 | 2 | 50.00 | 10 | 32.26 |
| Improved | 3 | 60.00 | 2 | 50.00 | 14 | 77.78 | 2 | 50.00 | 21 | 67.74 |
| **Low energy or tiredness** | | | | | | | | | | |
| Worsened | – | – | – | – | – | – | – | – | – | – |
| No change | 3 | 60.00 | 2 | 50.00 | 5 | 27.78 | 3 | 75.00 | 13 | 41.94 |
| Improved | 2 | 40.00 | 2 | 50.00 | 13 | 72.22 | 1 | 25.00 | 18 | 58.06 |
| **Muscle or body aches** | | | | | | | | | | |
| Worsened | – | – | – | – | 1 | 5.56 | – | – | 1 | 3.23 |
| No change | 2 | 40.00 | 2 | 50.00 | 6 | 33.33 | 3 | 75.00 | 13 | 41.94 |
| Improved | 3 | 60.00 | 2 | 50.00 | 11 | 61.11 | 1 | 25.00 | 17 | 54.84 |
| **Headache** | | | | | | | | | | |
| Worsened | – | – | – | – | – | – | – | – | – | – |
| No change | 2 | 40.00 | 2 | 50.00 | 12 | 66.67 | 3 | 75.00 | 19 | 61.29 |
| Improved | 3 | 60.00 | 2 | 50.00 | 6 | 33.33 | 1 | 25.00 | 12 | 38.71 |
| **Chills or shivering** | | | | | | | | | | |
| Worsened | – | – | – | – | 1 | 5.56 | – | – | 1 | 3.23 |
| No change | 3 | 60.00 | 3 | 75.00 | 7 | 38.89 | 1 | 25.00 | 14 | 45.16 |
| Improved | 2 | 40.00 | 1 | 25.00 | 10 | 55.56 | 3 | 75.00 | 16 | 51.61 |
| **Feeling hot or feverish** | | | | | | | | | | |
| Worsened | – | – | – | – | – | – | – | – | – | – |
| No change | 2 | 40.00 | 1 | 25.00 | 6 | 33.33 | 1 | 25.00 | 10 | 32.26 |
| Improved | 3 | 60.00 | 3 | 75.00 | 12 | 66.67 | 3 | 75.00 | 21 | 67.74 |
| **Nausea (feeling like you wanted to throw up)** | | | | | | | | | | |
| Worsened | – | – | – | – | – | – | – | – | – | – |
| No change | 5 | 100.00 | 4 | 100.00 | 13 | 72.22 | 4 | 100.00 | 26 | 83.87 |
| Improved | – | – | – | – | 5 | 27.78 | – | – | 5 | 16.13 |

**Table 4** (continued)

| | Day 7 − Follow-up | | | | | | | | | |
|---|---|---|---|---|---|---|---|---|---|---|
| | $1 \times 10^8$ (N = 5) | | $5 \times 10^8$ (N = 5) | | $1 \times 10^9$ (N = 20) | | $1 \times 10^{10}$ (N = 5) | | Total (N = 35) | |
| | N | % | N | % | N | % | N | % | N | % |
| How many times did you vomit (throw up) in the last 24 h? | | | | | | | | | | |
| Worsened | – | – | – | – | 1 | 5.56 | – | – | 1 | 3.23 |
| No change | 5 | 100.00 | 4 | 100.00 | 17 | 94.44 | 4 | 100.00 | 30 | 96.77 |
| Improved | – | – | – | – | 1 | 5.56 | – | – | 1 | 3.23 |
| B | | | | | | | | | | |
| How many times did you have diarrhea (loose or watery stools) in the last 24 h? | | | | | | | | | | |
| Worsened | – | – | – | – | – | – | – | – | – | – |
| No change | 5 | 100.00 | 3 | 75.00 | 10 | 55.56 | 3 | 75.00 | 21 | 67.74 |
| Improved | – | – | 1 | 25.00 | 7 | 38.89 | 1 | 25.00 | 9 | 29.03 |
| Rate your sense of smell in the last 24 h | | | | | | | | | | |
| Worsened | – | – | – | – | 1 | 5.56 | – | – | 1 | 3.23 |
| No change | 4 | 80.00 | 2 | 50.00 | 12 | 66.67 | 4 | 100.00 | 22 | 70.97 |
| Improved | 1 | 20.00 | 2 | 50.00 | 5 | 27.78 | – | – | 8 | 25.81 |
| Rate your sense of taste in the last 24 h | | | | | | | | | | |
| Worsened | – | – | – | – | 1 | 5.56 | 1 | 25.00 | 2 | 6.45 |
| No change | 4 | 80.00 | 2 | 50.00 | 10 | 55.56 | 3 | 75.00 | 19 | 61.29 |
| Improved | 1 | 20.00 | 2 | 50.00 | 7 | 38.89 | – | – | 10 | 32.26 |
| In the past 24 h, have you returned to your usual health (before your COVID-19 illness)? | | | | | | | | | | |
| Worsened | – | – | – | – | – | – | – | – | – | – |
| No change | 4 | 80.00 | 2 | 50.00 | 17 | 100.00 | 4 | 100.00 | 27 | 90.00 |
| Improved | 1 | 20.00 | 2 | 50.00 | – | – | – | – | 3 | 10.00 |
| In the past 24 h, have you returned to your usual activities (before your COVID-19 illness)? | | | | | | | | | | |
| Worsened | – | – | – | – | – | – | – | – | – | – |
| No change | 2 | 40.00 | 2 | 50.00 | 17 | 94.44 | 4 | 100.00 | 25 | 80.65 |
| Improved | 3 | 60.00 | 2 | 50.00 | 1 | 5.56 | – | – | 6 | 19.35 |
| In the past 24 h, what was the severity of your overall COVID-19-related symptoms at their worst? | | | | | | | | | | |
| Worsened | – | – | – | – | – | – | – | – | – | – |
| No change | 2 | 40.00 | – | – | 3 | 16.67 | 1 | 25.00 | 6 | 19.35 |
| Improved | 3 | 60.00 | 4 | 100.00 | 15 | 83.33 | 3 | 75.00 | 25 | 80.65 |

consent before enrollment. The experiments conformed to the principles set out in the WMA Declaration of Helsinki and the Department of Health and Human Services Belmont Report. Trial details, including inclusion and exclusion criteria, were deposited at ClinicalTrials.gov (NCT04747574) and can be accessed anytime. The authors vouch for the accuracy and completeness of the data and for the fidelity of the trial protocol.

### COVID-19 global severity and symptoms questionnaire

Patients were asked to complete the COVID-19 symptoms questionnaire at presentation and on Days 5 and 7. At screening, all patients completed the questionnaire. The most common symptoms present in at least two-thirds (66.7%) of these patients were shortness of breath (80%; 28 of 35), cough (80%; 28 of 35), low energy or tiredness (80%; 28 of 35), and muscle or body aches (68.6%; 24 of 35;

Table 4A and B). The patients ranked the severity of their symptoms on as either mild, moderate, or severe.

### Statistical analysis

The study was not expected to show statistical significance, but rather, to provide assessment for the primary and other end points. No official hypothesis testing was performed, and no statistical analysis plan was prepared. All data obtained in this study were listed and tabulated. Statistical analyses were performed using statistical software SAS® v9.4 (SAS Institute, Cary, NC, USA) and SPSS V28 (IBM, NYC). All statistical analyses are descriptive in nature. Continuous variables are summarized by a count, mean, standard deviation, minimum, median and maximum, and categorical variables by a count and proportion. Baseline values were defined as the last valid value prior to first study drug administration. Confidence

**The paper explained**

**Problem**

To date (June 22), 539 million COVID-19 cases were detected worldwide with over 6.32 million related deaths. 5% of COVID-19 patients develop a life-threatening lung disease.

**Results**

CD24 negatively regulates the immune system, by potent inhibition of NFκB activation. CD24 selectively suppresses host response to danger- but not to pathogen- associated molecular patterns, and hence, it inhibits the cytokine storm without inhibiting viral clearance. Technology was validated, for safety and efficacy, in a set of *in vitro* and *in vivo* studies. Very promising clinical efficacy has been shown, moderate-to-severe COVID-19 patients have been enrolled in phase Ib/IIa clinical trial. No drug-related adverse events were observed regardless of dose group.

**Impact**

EXO-CD24 treats the most important complication of COVID-19, the cytokine storm, which can lead to rapid clinical deterioration that might result in pulmonary insufficiency and even death. EXO-CD24 target only those 5% who needs therapy consequentially, the total number of COVID-19 patients that could be treated by EXO-CD24 is expected to reach 4.6 million patients by 2025. The same technology may also be applied to other diseases with hyperactive immune system and unmet therapeutic needs.

intervals, where relevant, are two-sided with a confidence level of 95%. Safety assessment was based on AEs. All AEs were coded using the Medical Dictionary for Regulatory Activities System Organ Class and Preferred Term.

Independent-samples *t*-test for continuous variables and chi-square tests for categorical variables was used to compare between the intervention group and the historical controls. All tests were two-tailed and $P < 0.05$ were considered significant.

## Data availability

This study includes no data deposited in external repositories.

**Expanded View** for this article is available online.

## Acknowledgements

We are indebted to Mrs. Ana Jukant Geralnik, an English-speaking professional, for editing and proofreading. This work was partial granted by the Israeli Innovation Authority (16501).

## Author contributions

**Shiran Shapira:** Conceptualization; data curation; analysis; investigation; methodology; project administration; writing—original draft; writing—review and editing. **Marina Ben Shimon:** Data curation and investigation. **Mori Hay-Levi:** Data curation and recruitment of patients. **Gil Shenberg:** Data curation and recruitment of patients. **Guy Choshen:** Data curation and recruitment of patients. **Lian Bannon:** Data curation and recruitment of patients. **Michael Tepper:** Conceptualization. **Dina Kazanov:** Data curation and investigation. **Jonathan Seni:** Writing—original draft; writing—review and editing. **Shahar Lev-Ari:** Data analysis; formal analysis. **Michael Peer:** Conceptualization. **Dimitrios Boubas:** Writing—review and editing. **Justin Stebbing:** Writing—review and editing. **Sotirios Tsiodras:** Conceptualization and Writing—review and editing. **Nadir Arber:** Conceptualization; supervision; methodology; investigation; funding acquisition; writing—original draft; writing—review and editing.

## Disclosure and competing interests statement

Prof. Arber, Dr. Marina Ben-Shimon and Dr. Shapira receive salary and have options in OBCTCD24 LTD.

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
