## [Review Process File · EMBO Molecular Medicine]

A novel platform for attenuating immune hyperactivity using EXO-CD24 in Covid-19 and beyond

Shiran Shapira, Marina Ben-Shimon, Mori Hay-Levi, Gil Shenberg, Guy Choshen, Lian Bannon, Michael Tepper, Dina Kazanov, Jonathan Seni, Shahar Lev-Ari, Michael Peer, Dimitrios Boubas, Justin Stebbing, Sotirios Tsiodras, and Nadir Arber

DOI: [10.15252/emmm.202215997](https://doi.org/10.15252/emmm.202215997)

Corresponding authors: Nadir Arber (nadira@tlvmc.gov.il)

Review Timeline:

Submission Date:	13th Mar 22
Editorial Decision:	13th Apr 22
Authors' Correspondence:	13th Apr 22
Editor's Correspondence:	14th Apr 22
Revision Received:	2nd May 22
Editorial Decision:	20th May 22
Revision Received:	27th May 22
Editorial Decision:	8th Jun 22
Revision Received:	23rd Jun 22
Accepted:	27th Jun 22

Editor: *Zeljko Durdevic*

Transaction Report:

13th Apr 2022

Dear Prof. Arber,

Thank you for the submission of your manuscript to EMBO Molecular Medicine. We have now received feedback from the three reviewers who agreed to evaluate your manuscript. As you will see from their reports pasted below, While referee #1 is overall positive, referees #2 and #3 recognize interest of the clinical study but also raise serious and partially overlapping concerns. Given the nature of these criticisms, addressing all the referees' comments would require a lot of additional work, time and effort. Given these considerations, I am afraid that we do not feel it would be productive to call for a revised version of your manuscript at this stage and therefore we cannot offer to publish it.

Given the potential interest of the findings, we would, however, be willing to consider a new manuscript on the same topic if at some time in the near future you obtained data that would considerably strengthen the message of the study and address the referees concerns in full. Particular attention should be given to providing following data:

- Vaccination status of the participants.
- Historical cohort should be better defined i.e. standard of care, vaccination status etc. Alternatively refocus the study on the safety and tolerability, rather than as an efficacy study with a suboptimal control cohort.
- Add pharmacokinetic data either in patients or in an animal model.
- Add preclinical data in an appropriate animal model.
- Provide more details on exosome generation.
- Re-evaluate statistical analyses.

To be completely clear, however, I would like to stress that if you were to send a new manuscript this would be treated as a new submission rather than a revision and would be reviewed afresh, in particular with respect to the literature and the novelty of your findings at the time of resubmission. If you decide to follow this route, please make sure you nevertheless upload a letter of response to the referees' comments.

I am sorry that I could not bring better news this time and hope that the referee comments are helpful in your continued work in this area.

Yours sincerely,

Zeljko Durdevic

***** Reviewer's comments *****

Referee #1 (Remarks for Author):

In this work the authors describe an innovative clinical trial for the treatment of COVID19 patients. The main goal of treatment is to treat the cytokine storm. For this purpose, they recruit the CD24 molecule that has been shown in various studies, including by Arber et al. From these previous findings it has been suggested that CD24 may suppress the severe inflammation via its immunomodulation.

One of the challenges in treating with proteins or cytokines is the problem of delivery to the target site. In this work, the authors chose to genetically express CD24 on the exosome. Inhalation of the exosomes led the CD24 protein directly to the lungs and thus created a high local concentration.

Thus, it seems that the use of the new EXO-CD24 treatment in COVID19 patients a rapid decrease in inflammatory markers and cytokine / chemokine levels. Compared with historical controls, the length of hospitalization, intensive care hospitalizations, mechanical respiration, and mortality rates were shorter ($p < 0.05$). Special efforts were invested to examine the safety and follow the side effects. Patients were added to the trial only after the doses that were given appeared to be safe. Indeed, the safety results look promising.

The authors also argue that the system they have developed can serve as a therapeutic platform for other lung and systemic diseases characterized by cytokine outbreaks.

The study is important and ready for publication.

At the same time there are a number of comments that I think will contribute to the quality of the article.

Line 28: Exosomes or EV ? The preferred expression is EV, see quotes below. Exosomes can be used, but it should be clarified that a thorough examination has not been performed to make sure that they are exosomes and no other EVs. The name EXO-CD24 can in my opinion be left anyway.

"Without doubt, MISEV2018 represents the consensus of by far the largest group of EV experts assembled to date as an author team, and in this sense, "extracellular vesicle" is the expert consensus for the general term. Of course, this undisputed fact must be balanced against the many authors who use different definitions of "exosome" individually."

[1] Théry C, Witwer KW, Aikawa E, et al. Minimal information for studies of extracellular vesicles 2018 (MISEV2018): a position statement of the International Society for Extracellular Vesicles and update of the MISEV2014 guidelines. *J Extracell Vesicles*. 2018;7(1):1535750 Available from: <https://www.tandfonline.com/doi/full/10.1080/20013078.2018.1535750> [PMC free article] [PubMed] [Google Scholar]

Line 94: It is not clear how the exosomes were isolated. There is no detail and no reference.

Line 97: In what volume is the treatment given

Line 101: Is there any pre-clinical data on Exo-CD24?

Line 102: The sentence is not clear enough.

Line 105: Will be or was ?

Line 352: Enthusiastic and unscientific expressions should be avoided (pleasant, game changer).

Referee #2 (Remarks for Author):

This is a report of a clinical study conducted in the midst of the pandemic on an intriguing therapeutic candidate. Of course the major limitation is the lack of a real control group, as acknowledged by the authors. There do not appear to have been significant SAEs, which should support placebo-controlled studies (including in indications beyond COVID) going forward. That said, there are several items that could improve the manuscript if addressed:

- 1) How was the dose derived?
- 2) Could the authors provide a PAGE of exo-24 prep? Is there a placebo or control preparation of exosomes?
- 3) Please include more details of the background preclinical data in support of this product. Was exo-24 administered by similar nebulization in animal models where direct target engagement was assessed?
- 4) Are there any biomarkers from preclinical work that can be followed in patients to assess in vivo target engagement?
- 5) Is exo-24 detectable systemically following nebulization?
- 6) Was standard of care different during the study vs. historical controls? It would be nice to have some control patients from after the study enrollment to ensure improvements in outcomes in the study group were exo-24 related, and not a reflection of improvements in standard of care as the pandemic has progressed.
- 7) Were cytokine measurements performed on blood or BAL samples?
- 8) In the absence of a placebo group, it would be nice to see some dose-dependent effects. Why are so few patients from groups 1 and 2 included in Fig. 5? This seems like a great opportunity to demonstrate a dose-dependent effect.
- 9) What is meant by "investigations" as an AE?

Referee #3 (Comments on Novelty/Model System for Author):

Phase I study. Thus there is no model system. Intervention was used in COVID-19 patients. However, historical cohort for control is ill defined.

Referee #3 (Remarks for Author):

In the present study, Shapira and colleagues demonstrate clinical efficacy of a novel CD24 inhibitor, EXO-CD24, in patients with moderate-high severity COVID-19. They noted absence of adverse effects and a reduction in inflammatory markers after administration of EXO-CD24. When comparing clinical values to historical data from Covid-19 patients, they detected significantly shorter duration of hospitalization, admissions to ICU, mechanical ventilation and mortality rates.

Despite reduced percentages of SARS-CoV2-related hospitalization due to vaccination efforts and emergence of new virus variants, Covid-19 remains a life-threatening disease especially to patients with impaired immune system. Therefore, effective therapies are urgently required. The present study describes that EXO-CD24 is tolerable even at high doses, which may justify further clinical investigation of this substance. My major concern with this study is the prominent display of clinical data that

may or may not have improved due to the intervention. The study is not at all designed to evaluate clinical improvement in comparison to a very ill defined "historical cohort". This needs to be toned down massively or even deleted. An alternative would be to provide a lot more clinical data on patients enrolled AND the historical cohort.

The key point of this phase I study is safety and this should be clarified and displayed accordingly. A typical phase I study includes (in addition to safety data) pharmacokinetic data. I understand that this can be challenging in a drug that needs to be inhaled. However, it is possible and inhaled drugs also need proper evaluation for distribution and drug levels in the target organ and systemically. This is completely missing. Has this been performed? Even animal data would be helpful to assess this issue. In addition, CD24 levels could have been assessed in BALF or sputum of patients. Or even in plasma (increase after application). Otherwise it is completely unknown whether the observed effects are based on the drug or whether the observed effects are the normal clinical course of the disease (see comments below).

Major comments:

1. The title should clarify that this is a phase I clinical trial. Title right now reads like a review article.
2. Line 89: there is far too little information provided on how exosomes were generated. The cited literature (Shapira et al.) provides information on an expression system. But not on how CD24 is packaged into exosomes. This needs to be clarified so that others can use the system.

3. Clinical data presentation, cytokine levels, radiography: this is presented very prominently and in comparison to a not well defined "historical cohort", line 181.

During the study period (what was the exact period of enrollment? Not provided!) Israel massively ramped up vaccination in its population. There are no data indicating whether study participants got vaccinated during or before the study. What about the historical cohort with regard to vaccination? Remdesivir was given to 57% of the EXO group, but only to 24% of the historical group. Why? I doubt that this is not a statistically significant difference. It is only table 5 and table 6 that compare with the historical cohort. Figures 2-5 compare baseline data to later time-points in the intervention group only. Why not comparing the data to the ones of the control group? This would probably show the normal course of cytokine levels of patients receiving remdesivir and dexamethasone without any difference to the intervention group. The way the data are presented is very suggestive for an effect produced by the intervention - however, this is false interpretation of the data. In its current form I cannot support publication of the data in a high ranking journal such as EMBO MM.

4. There is far too little information provided in the figure legends. N, statistical evaluation. Why were only some patients evaluated and not all (Fig. 4 and 5). Why presenting X-rays of a single patient (Fig. 2)? Doesn't provide any useful information.

5. The abstract claims that the intervention led to significantly reduced mortality (line 34) with a p-Value of <0.05. However, table 6 reads very differently - no significant difference. Here: "Alive (%)" no significant difference. This is a major error! The description of the table with regard to mortality is completely missing. What does "alive" mean?

6. The same misleading statements are made in the discussion line 356: "C). a statistically significant improvement in the length of hospitalization, transfer to ICU, need for invasive ventilation support and even mortality."

Where is the data for "transfer to ICU", "ventilation and mortality"? Is there a supplement missing? What has been the group for comparison? Table 6 and other tables are not providing these data!

7. The discussion needs to be rewritten, as it is very repetitive and largely repeating the findings from the result section. Authors should use more objective/ scientific and less "advertising" language (e.g. lines 347, 399, "game changer"). In the current phase of clinical development this intervention is clearly not yet a game changer.

8. Are there pharmacokinetic data? This is a major goal of phase I studies apart from generating safety data. Please include if present or discuss why lacking.

9. Line 375: a lot more statements that are not backed by data or citations.

10. Line 395: citations for the many preclinical studies missing!

11. Generally, the manuscript misses an important link to preclinical data showing the efficacy of EXO-CD24 in vitro and in vivo. If these have not been generated at all, it may even be ethically questionable to perform a phase I trial in COVID-19 patients.

12. Line 393: CD24 is NOT selective for COVID-19 induced hyperinflammation.

13. All statistics need to be re-evaluated by a clinical statistician.

14. There is virological data missing. What was the dominant SARS-CoV-2 variant at the time. Differences in comparison with the historical cohort?

Minor comments:

1. The manuscript contains several typos and grammar issues that should be corrected. Examples are lines 67 (full stop missing), 71 (comma missing), line 407 "in".

2. Line 72 on DAMPS versus PAMPS. This statement needs a citation.

As a service to authors, EMBO provides authors with the possibility to transfer a manuscript that one journal cannot offer to publish to another EMBO publication. The full manuscript and if applicable, reviewers reports are automatically sent to the receiving journal to allow for fast handling and a prompt decision on your manuscript. For more details of this service, and to transfer your manuscript to another EMBO title please click on Link Not Available

Dear editor,

Thank you for your letter of today.

I appreciate the openness and sincerity

Luckily WE DO HAVE ANSWERS to all of the reviewers comments !!!

We had previously submitted, to Nature, a combine paper of the pre-clinical and clinical data

One of their major comment was that we should separate the pre-clinical and clinical data

In order to improve the paper this is what we had done in the current submission

With your permission we can re-submit the paper addressing all the constructive comments of the reviewers

In fact we do agree that this is a better way to present our breakthrough novel technology

We can send a revise manuscript in 10-14 days

Please approve

Dear Prof. Arber,

thank you for your e-mail. We would welcome submission of the revised manuscript, which would include pre-clinical data and in which all referees' concerns would be appropriately addressed.

Please consider changing the title that should be more informative and supported by the data presented. I look forward to reading your revised manuscript.

Best wishes,
Zeljko Durdevic

Referee #1 (Remarks for Author):

In this work the authors describe an innovative clinical trial for the treatment of COVID19 patients. The main goal of treatment is to treat the cytokine storm. For this purpose, they recruit the CD24 molecule that has been shown in various studies, including by Arber et al. From these previous findings it has been suggested that CD24 may suppress the severe inflammation via its immunomodulation.

One of the challenges in treating with proteins or cytokines is the problem of delivery to the target site. In this work, the authors chose to genetically express CD24 on the exosome. Inhalation of the exosomes led the CD24 protein directly to the lungs and thus created a high local concentration. Thus, it seems that the use of the new EXO-CD24 treatment in COVID19 patients a rapid decrease in inflammatory markers and cytokine / chemokine levels. Compared with historical controls, the length of hospitalization, intensive care hospitalizations, mechanical respiration, and mortality rates were shorter ($p < 0.05$). Special efforts were invested to examine the safety and follow the side effects. Patients were added to the trial only after the doses that were given appeared to be safe. Indeed, the safety results look promising.

The authors also argue that the system they have developed can serve as a therapeutic platform for other lung and systemic diseases characterized by cytokine outbreaks.

The study is important and ready for publication.

At the same time there are a number of comments that I think will contribute to the quality of the article.

Line 28: Exosomes or EV ? The preferred expression is EV, see quotes below. Exosomes can be used, but it should be clarified that a thorough examination has not been performed to make sure that they are exosomes and no other EVs. The name EXO-CD24 can in my opinion be left anyway. "Without doubt, MISEV2018 represents the consensus of by far the largest group of EV experts assembled to date as an author team, and in this sense, "extracellular vesicle" is the expert consensus for the general term. Of course, this undisputed fact must be balanced against the many authors who use different definitions of "exosome" individually."

[1] Théry C, Witwer KW, Aikawa E, et al. Minimal information for studies of extracellular vesicles 2018 (MISEV2018): a position statement of the International Society for Extracellular Vesicles and update of the MISEV2014 guidelines. *J Extracell Vesicles.* 2018;7(1):1535750 Available from: <https://www.tandfonline.com/doi/full/10.1080/20013078.2018.1535750>

[PMC free article] [PubMed] [Google Scholar]

Response:

The reviewer is absolutely right. If you wish to change "exosomes" to the generic term "EV" it can be done, although we prefer, as the reviewer suggested, to "The name EXO-CD24 can in my opinion be left anyway". Currently, there are three primary classifications of extracellular vesicles; apoptotic bodies, microvesicles (known as ectosomes, microparticles, or shedding vesicles), and exosomes. In

contrast to not knowing the whole population's composition, we are certain that exosomes are included since a single vesicle, stained twice in flow cytometry, shows that the same particle that expresses CD24 also expresses a biomarker for exosomes such as CD81. Considering the method of processing and the size distribution of the particles, we are sure the product does not contain apoptotic bodies.

Line 94: It is not clear how the exosomes were isolated. There is no detail and no reference.

Response:

The preparation of EXO-CD24 is described in a paragraph in the section "Preparation of CD24 displaying exosomes" (page 9 in the track changes (TC) version).

Line 97: In what volume is the treatment given

Response:

Prior to administration, 1.5 mL saline (0.9% sodium chloride Intravenous Infusion BP) are added, and the total volume of suspension is transferred to the nebulizer chamber, standard hospital-grade inhalation device, for inhalation and administered once daily (QD) for 5 days. This description was added to page 14 in the TC version)

Line 101: Is there any pre-clinical data on Exo-CD24?

Response:

Extensive preclinical data (in vitro and in vivo) regarding EXO-CD24 was added.

- ***EXO-CD24 production and characterization (pages 8-10, 21 in the TC version)***
- ***In vitro cytokine/chemokine secretion studies (pages 11 and 22 in the TC version)***
- ***In vivo studies (pages 11):***
 - ***Dose toxicity study (pages 11-12, 22-23 in the TC version)***
 - ***LPS-induced ARDS model (pages 12 and 23-24 in the TC version)***
 - ***Cecum Ligation and Puncture (CLP) induced sepsis model (pages 12-13, 24 in the TC version)***
 - ***PK study (pages 13 and 24 in the TC version)***

Line 102: The sentence is not clear enough.

Response:

Thanks for the comment. The sentence was re-written:

"Analysis and examination of EXO-CD24 secreting cells (Trex293-CD24) and the resulting investigational drug substrate and drug product were characterized and tested, for safety and efficacy. Many different analytical and biological tests were conducted including

identity, purity, content, potency, microbial (mycoplasma, sterility and endotoxin LAL) and viral contamination validation tests" (page 14 in the TC version)

Line 105: Will be or was ?

Response:

The typo was corrected to "was". (line 266 in the TC version)

Line 352: Enthusiastic and unscientific expressions should be avoided (pleasant, game changer).

Response:

We agree with the reviewer comment and have accordingly tone down the Enthusiastic and unscientific expressions

Referee

#2

:

1) How was the dose derived?

Response:

We thank the reviewer and have clarified this in the new much changed manuscript. The initial dose (1×10^8 particles) was based on similar studies using viruses as the therapeutic vehicle, as well as on

a study aimed at evaluating the safety and tolerance of inhaled mesenchymal-cell-derived exosomes in healthy volunteers (NCT04313647). In addition, each one carries 10^8 exosomes, hence we decided to initiate our phase 1 study with this dosage, that was increase gradually to 5×10^8 , 10^9 and even 10^{10} exosomes as we had not seen any SAE and even AE. In parallel these dosages were found to be very effective in pre-clinical studies.

2) Could the authors provide a PAGE of exo-24 prep? Is there a placebo or control preparation of exosomes?

Response

The preparation of EXO-CD24 is described in the section "Preparation of CD24 displaying exosomes" (page 9 in the TC version). If the author request, we can elaborate more. In addition, CD24-negative exosomes were prepared using the same method and protocol from the same cells. The only difference being that they had not been inducted with tetracycline. Non inducible exosomes were used in several in vitro and in vivo studies. In the clinical trials, as instructed by the regulatory agencies and our consultants saline was used as the placebo.

3) Please include more details of the background preclinical data in support of this product. Was exo-24 administered by similar nebulization in animal models where direct target engagement was assessed?

Response

Thanks for asking. More details regarding the preclinical data, in vitro and in vivo, were added, including production, characterization, safety and efficacy studies (pages 8-13 and 21-24 in the TC pages). It was originally suggested to us to separate the pre-clinical and the clinical data. In light of the reviewers' constructive comments, we understand and agree that it is better to consolidate the data into one paper. EXO-CD24 was administrated to animals by inhalation as aerosol vapors, similar to the way it is given to humans, using an inhalation cage.

4) Are there any biomarkers from preclinical work that can be followed in patients to assess in vivo target engagement?

Response

An important point that now is better addressed. Blood samples had been collected from all patients during the entire study. Aliquots were tested for cytokine arrays in order to compare the clinical response to the systemic level of cytokines/chemokines. EXO-CD24 are of human origin, hence biomarkers cannot distinguish them from endogenous CD24 as in each ml of blood there are 10^8 exosomes and they express the same biomarkers. However, in vivo after inhalation, the exosomes were detected in the BALF of the treated animals meaning that they had reached their target organ, e.g the

lungs. Furthermore, we are currently applying a real-scale 3D model of patient's CT-derived airways. This model, which was previously used for in-vitro aerosol studies, is adapted for the evaluation of EXO-CD24 delivery to the lungs by inhalation.

5) Is exo-24 detectable systemically following nebulization?

Response

Thank you for raising this important point. According to the PK/PD studies after a single inhalation, EXO-CD24 was detectable not only in the serum but even in the bloodstream, as demonstrated by flow cytometry. (pages 13 and 24 in the TC version)

6) Was standard of care different during the study vs. historical controls? It would be nice to have some control patients from after the study enrollment to ensure improvements in outcomes in the study group were exo-24 related, and not a reflection of improvements in standard of care as the pandemic has progressed.

Response:

The Standard of care was fairly consistent throughout the study. All patients had received dexamethasone, clexane and nexuim. Many had received remdesivir (in the window of 10 days), and a few had received Actemra (tocilizumab). A detailed list was included in table 1. However, we agree with the reviewers' comment and have

accordingly decided to delete the historical controls from our manuscript.

7) Were cytokine measurements performed on blood or BAL samples?

Response:

Cytokines and chemokines were measured in the blood and BAL samples of mice, in order to demonstrate the efficacy of the suggested drug on systemic and local inflammatory indices.

In humans, it was not ethically to perform bronchoscopy in these severe ill patients; hence the level of cytokine and chemokines was measured only in the blood.

8) In the absence of a placebo group, it would be nice to see some dose-dependent effects. Why are so few patients from groups 1 and 2 included in Fig. 5? This seems like a great opportunity to demonstrate a dose-dependent effect.

Response:

We believe that EXO-CD24 is very effective. The lower dose might be as effective as the high dose. The clinical study did not find any significant differences in clinical (oxygen saturation and respiratory rate) and laboratory (CRP, ferritin, fibrinogen, LDH, NLR) indices,

between the different dose groups, as well as in the measured levels of cytokines and chemokines.

9) What is meant by "investigations" as an AE?

Response

Thank you for raising the point. Now it is better explained in the text. Any complaint of a patient or lab abnormality was scored and regarded as AE. 80% had at least one AE however none was related to the drug. SAE was reported in 2.8% but again none was related to the drug

Referee #3 (Comments on Novelty/Model System for Author):

Phase I study. Thus there is no model system. Intervention was used in COVID-19 patients. However, historical cohort for control is ill defined.

Response

Thank you for raising this important issue. In phase Ib/2a studies it is common to include historical controls in order to hint for efficacy. It is especially true in the Covid19 era. We tried to match the historical controls as much as possible. Three controls for each participant match for age, gender, time of hospitalization, clinical parameters and standard of therapy. However, based on the editor's

request this paragraph was deleted from the manuscript.

In the present study, Shapira and colleagues demonstrate clinical efficacy of a novel CD24 inhibitor, EXO-CD24, in patients with moderate-high severity COVID-19. They noted absence of adverse effects and a reduction in inflammatory markers after administration of EXO-CD24. When comparing clinical values to historical data from Covid-19 patients, they detected significantly shorter duration of hospitalization, admissions to ICU, mechanical ventilation and mortality rates.

Despite reduced percentages of SARS-CoV2-related hospitalization due to vaccination efforts and emergence of new virus variants, Covid-19 remains a life-threatening disease especially to patients with impaired immune system. Therefore, effective therapies are urgently required. The present study describes that EXO-CD24 is tolerable even at high doses, which may justify further clinical investigation of this substance. My major concern with this study is the prominent display of clinical data that may or may not have improved due to the intervention. The study is not at all designed to evaluate clinical improvement in comparison to a very ill defined "historical cohort". This needs to be toned down massively or even deleted. An alternative would be to provide a lot more clinical data on patients enrolled AND the historical cohort.

The key point of this phase I study is safety and this should be clarified and displayed accordingly. A typical phase I study includes (in addition to safety data) pharmacokinetic data. I understand that this can be challenging in a

drug that needs to be inhaled. However, it is possible and inhaled drugs also need proper evaluation for distribution and drug levels in the target organ and systemically. This is completely missing. Has this been performed? Even animal data would be helpful to assess this issue. In addition, CD24 levels could have been assessed in BALF or sputum of patients. Or even in plasma (increase after application). Otherwise it is completely unknown whether the observed effects are based on the drug or whether the observed effects are the normal clinical course of the disease (see comments below).

Response

We absolutely agree with the reviewer and his comments are well taken. There is no doubt that Phase I is intended to evaluate human safety first and foremost. The Israeli MOH had not granted a permission to include a placebo arm in this study. As we have a lot of experience in the treatment of Covid19 patients it was pleasant to sense the possible and potential efficacy of EXO-CD24. To that extent, we had carefully searched for comparable historical controls and the subjective reports of the patients (accepted in Covid19 studies). According to the editor's request and most probably the current reviewer we had omitted the comparison to historical control and focused on incorporating preclinical information and providing the information regarding the safety of the product in humans.

The efficacy of EXO-CD24 was tested and verified in vitro by adding monoclonal antibodies to CD24 in order to inhibit or block the

activity of the drug. Indeed, the effect of EXO-CD24 on cytokine secretion was partially but significantly blocked

Major

comments:

1. The title should clarify that this is a phase I clinical trial. Title right now reads like a review article.

Response

An important comment. The title has been changed accordingly

2. Line 89: there is far too little information provided on how exosomes were generated. The cited literature (Shapira et al.) provides information on an expression system. But not on how CD24 is packaged into exosomes. This needs to be clarified so that others can use the system.

Response

The preparation of EXO-CD24 is described in the section "Preparation of CD24 displaying exosomes" (page 9 in the TC version). If the reviewer finds it deems suitable we can add more details.

3. Clinical data presentation, cytokine levels, radiography: this is presented very prominently and in comparison to a not well defined "historical cohort",
line 181.

During the study period (what was the exact period of enrollment? Not provided!) Israel massively ramped up vaccination in its population. There are no data indicating whether study participants got vaccinated during or before the study. What about the historical cohort with regard to vaccination?

Remdesivir was given to 57% of the EXO group, but only to 24% of the historical group. Why? I doubt that this is not a statistically significant difference. It is only table 5 and table 6 that compare with the historical cohort. Figures 2-5 compare baseline data to later time-points in the intervention group only. Why not comparing the data to the ones of the control group? This would probably show the normal course of cytokine levels of patients receiving remdesivir and dexamethasone without any difference to the intervention group. The way the data are presented is very suggestive for an effect produced by the intervention - however, this is false interpretation of the data. In its current form I cannot support publication of the data in a high ranking journal such as EMBO MM.

Response

On Sep. 24, 2020, Israel's MoH approved the trial at TASMC. The first patient was recruited on September 26th (page 23 line 489 in the TC version). The last patient was enrolled in February 13th, 2021 (page 24 line 490 in the TC version). Vaccination was initiated in Israel in late December 2020. None of our patients, nor those of the historical controls, had been vaccinated. As mentioned, we accept the comments of the editor and reviewer. The section on historical controls was omitted from the paper

4. There is far too little information provided in the figure legends. N, statistical evaluation. Why were only some patients evaluated and not all (Fig. 4 and 5). Why presenting X-rays of a single patient (Fig. 2)? Doesn't provide any useful information.

Response

Thanks for the comment. The figure legends have been modified and clarified. At the time of submission, the first analyses were not performed on the entire cohort of patients, however, they have been performed and completed later. They are fully described in figure 10C. The X-ray is just a proof of potential efficacy that we had seen in most of the patients. If the reviewer demands it can definitely be deleted

5. The abstract claims that the intervention led to significantly reduced mortality (line 34) with a p-Value of <0.05 . However, table 6 reads very differently - no significant difference. Here: "Alive (%)" no significant difference. This is a major error! The description of the table with regard to mortality is completely missing. What does "alive" mean?

Response

The reviewer is obviously right. Sorry for the typo. We agree that it is not needed in this paper, and the paragraph was deleted. We can

provide explanations directly to the reviewer outside, or within the paper if the editor and reviewer will decide to include this section of the historical controls.

6. The same misleading statements are made in the discussion line 356: "C).

Response

It is corrected now.

a statistically significant improvement in the length of hospitalization, transfer to ICU, need for invasive ventilation support and even mortality." Where is the data for "transfer to ICU", "ventilation and mortality"? Is there a supplement missing? What has been the group for comparison? Table 6 and other tables are not providing these data!

Response

My apologies for the omission. If we keep this session, we will provide the data, but currently this section has been omitted based on the editor's and reviewer's request.

7. The discussion needs to be rewritten, as it is very repetitive and largely repeating the findings from the result section. Authors should use more objective/ scientific and less "advertising" language (e.g. lines 347, 399,

"game changer"). In the current phase of clinical development this intervention is clearly not yet a game changer.

Response

Indeed we are excited, but science is science. The reviewer comments are well taken. The "discussion" have been modified, tone down and rewritten according to the reviewer's suggestion.

8. Are there pharmacokinetic data? This is a major goal of phase I studies apart from generating safety data. Please include if present or discuss why lacking.

Response

We do not have pharmacokinetic data from patients. It is quite difficult, and most probably impossible, to distinguish between natural exosomes found within the body that display the exosomal biomarkers from those that are derived from human cells and produced in the laboratory that display the same biomarkers.

9. Line 375: a lot more statements that are not backed by data or citations.

Response

The statements were modified.

10. Line 395: citations for the many preclinical studies missing!

Response

More details regarding the preclinical data, in vitro and in vivo, were added.

11. Generally, the manuscript misses an important link to preclinical data showing the efficacy of EXO-CD24 in vitro and in vivo. If these have not been generated at all, it may even be ethically questionable to perform a phase I trial in COVID-19 patients.

Response

A significant amount of pre-clinical work have been conducted. EXO-CD24 is fully characterized, in vitro and in vivo. On top of many R&D batches, two engineering runs and more than five clinical batches had been produced and fully characterized. Each of the clinical batches is tested for safety and efficacy using many different analytical and biological tests. It includes various in process control tests, analysis of the drug substance and the drug product by different methods, ELISA, NTA, flow cytometry, Cryo-TEM, Western blot, biological activity, virology tests etc.

- ***EXO-CD24 production and characterization (pages 8-10, 21 in the TC version)***

- ***In vitro cytokine/chemokine secretion studies (pages 11 and 22 in the TC version)***
- ***In vivo studies (pages 11):***
 - ***Dose toxicity study (pages 11-12, 22-23 in the TC version)***
 - ***LPS-induced ARDS model (pages 12 and 23-24 in the TC version)***
 - ***Cecum Ligation and Puncture (CLP) induced sepsis model (pages 12-13, 24 in the TC version)***
 - ***PK study (pages 13 and 24 in the TC version)***

12. Line 393: CD24 is NOT selective for COVID-19 induced hyperinflammation.

Response

This sentence was deleted. However, we would like to emphasize that the treatment we offer is not limited to COVID19, but it can also be used to treat a wide range of respiratory and systemic diseases with a high risk for cytokine storm development. We are not targeting the virus itself but rather the most severe complication caused by the disease, the cytokine storm. Therefore, it is a platform that may be used for a variety of indications with unmet need.

Furthermore, we aim to provide treatment only to those 5% who truly need it, not to those who will recover on their own and only require supportive care.

13. All statistics need to be re-evaluated by a clinical statistician.

Response

Clinical statisticians, external and independent do the statistics. In addition, CSR was provided upon completion of the study.

14. There is virological data missing. What was the dominant SARS-CoV-2 variant at the time. Differences in comparison with the historical cohort?

Response

At the time of the study, the dominant SARS-CoV-2 variant in Israel was the alpha variant exactly like the control group

Minor

comments:

1. The manuscript contains several typos and grammar issues that should be corrected. Examples are lines 67 (full stop missing), 71 (comma missing), line 407 "in".

Response

I apologize for the typos. The manuscript has now been edited and proofread by an English-speaking professional.

2. Line 72 on DAMPS versus PAMPS. This statement needs a citation.

Response

Done

(reference

No.12)

20th May 2022

Dear Prof. Arber,

Thank you for the submission of your manuscript to EMBO Molecular Medicine. We have now received feedback from the three reviewers who agreed to evaluate your manuscript. As you will see from the reports below, while the referees #1 and #2 are supporting publication of the study, referee #3 acknowledges improvements of your revised manuscript but also raises concerns particularly regarding the quality of the figures and missing information about the replicates and statistics. Please address these points by improving the quality of the figures, providing more information about the number and nature of replicates and performing appropriate statistical analyses. Please show raw values and avoid statistical analyses when $n=2$. Please check our Author Guidelines:

<https://www.embopress.org/page/journal/17574684/authorguide#statisticalanalysis>

All other points regarding the mouse model and dosing should be addressed in writing and appropriately discussed in the manuscript.

EMBO Molecular Medicine encourages a single round of revision only and therefore, acceptance or rejection of the manuscript will depend on the completeness of your responses included in the next, final version of the manuscript. For this reason, and to save you from any frustrations in the end, I would strongly advise against returning an incomplete revision.

We would welcome the submission of a revised version within three months for further consideration. Please let us know if you require longer to complete the revision.

I look forward to receiving your revised manuscript.

Yours sincerely,

Zeljko Durdevic

*

**** Reviewer's comments ****

Referee #1 (Remarks for Author):

The authors have answered all my questions adequately and I have no further comments.

Referee #2 (Remarks for Author):

This is a significantly improved manuscript addressing key items from the initial review.

Referee #3 (Comments on Novelty/Model System for Author):

Novel manuscript provides a lot more in vivo mouse data.

A SARS-CoV-2 mouse infection model would be the correct model for the anticipated application. ARDS model may be suitable as well.

Referee #3 (Remarks for Author):

The revised manuscript was updated with a whole new dataset giving more information regarding Exosome production, in vivo efficacy, PK.

The authors followed the advise to delete the preliminary efficacy data and now rather focus on preclinical efficacy data and human safety data.

I still have comments:

1. quality of figures is sometimes very low. Fig. 1A: is this a paperscan?
2. I don't understand why the in vitro data were not repeated several times as it should be performed in accordance with good scientific practice. E.g. fig. 2: "Data shown is the average of duplicates from a single experiment." This is not the way data should be presented.
3. fig 5C, 2B and many other data: Statistics are missing.
4. Fig. 7, number of animals? Statistics?
5. The better animal model would have been SARS-CoV-2 infected mice. LPS induced lung injury may not be the proper model. Please discuss.
6. Fig. 6. in this waterfall plot, is there any statistical evaluation showing that survival is increased in the treatment group?
7. in the mouse models it seems like at least $10E9$ particles are required to achieve an effect. This was $10E9$ per mouse. $10E9$ was also given to humans. However, the human lung is much bigger than an animal lung. This suggests that this dose would not be sufficient to achieve an effect in humans. Even $10E10$ in humans would be far away and much lower compared to the $10E9$ that were given to mice. Please comment
8. given the different dosing groups in humans, was there a dose dependent effect in the inflammatory parameters. It is not clear which group is displayed here.

Dear Dr. Durdevic

Editor, EMBO

We are grateful for the opportunity to re-submit our manuscript "A novel platform for attenuating immune hyperactivation using exosomes displaying CD24 (EXO-CD24) in Coronavirus Disease-19 and beyond". We thank referees #1 and #2 for their support in publishing our study, as well as to referee #3 for her/his constructive comments

As requested, all points regarding the mouse model and dosing were addressed in writing and discussed in the manuscript.

Enclosed is the revised manuscript, modified according to the reviewer's comments. We have addressed each of the points and provided a point-by-point response that includes the comment followed by the response (italic). Changes in the manuscript text were made, according to the reviewers' comments, in track changes.

#Reviewer 3:

1. Quality of figures is sometimes very low. Fig. 1A: is this a paperscan?

The images were created with an old generation device. We have replaced it with an image from a newer device. Having said that, this figure is not critical and if it still fails to meet the high standard of EMBO Figure 1A can be deleted

2. I don't understand why the in vitro data were not repeated several times as it should be performed in accordance with good scientific practice. E.g. fig. 2: "Data shown is the average of duplicates from a single experiment." This is not the way data should be presented.

I apologize for the misunderstanding. To clarify, the experiment was conducted several times. Figure 2 is just a representative experiment as is a common practice in many articles.

3. fig 5C, 2B and many other data: Statistics are missing.

Figure 2B illustrates a representative experiment. Statistical data regarding the standard curves for each analyte are described below.

Analyte RANTES- Chi=2.24%, CV=0.36%, R²=1, DC=(65.47, 952172)

Analyte IL-1b- Chi=0.41%, CV=, R²=1, DC=(5.62, 14534)

Analyte CD40- Chi=5.08%, CV=1.93%, R²=0.999, DC=(139.27, 336912)

Analyte IL-1a- Chi=4.48%, CV=0.65%, R²=1, DC=(5.03, 7262)

Analyte IL-6- Chi=0.61%, CV=0.15%, R²=1, DC=(1.43, 44897)

Analyte MCP-1- Chi=3.40%, CV=0.36%, R²=1, DC=(1.61, 8149)

Mip-3a- Chi=2.19%, CV=0.065%, R²=1, DC=(2.71, 2104)

IL-10- Chi=0.59%, CV=0.014%, R²=1, DC=(16.57, 47726)

Notes:

CV-The Coefficient of Variation of the standard curve replicates at each dilution level.

Chi-The Chi-Square test statistic of the distance between observed concentrations with expected concentrations.

Indeed, in Figure 4A the SEM and the error bars were missing and were added.

In the figure legend 5C, , data is represented as average \pm SEM.

In each group, n=10 .

In Figure 10D the SD and the error bars were missing and were added.

4. Fig. 7, number of animals? Statistics?

A total of 132 mice were used. A total of 8 time points (20 minutes, 60 minutes, 4 hours, 8 hours, 12 hours, 24 hours, 48 hours, and 72 hours) were performed. Four mice in each group. The SEM was calculated and error bars were added to each graph. T-tests were calculated and the difference between EXO-CD24 and saline was statistically significant at time points between 60min-48h ($P<0.05$).

5. The better animal model would have been SARS-CoV-2 infected mice. LPS induced lung injury may not be the proper model. Please discuss.

EXO-CD24 is indicated for the treatment of ARDS. Covid19 is a private case of the ARDS. We had used the COVID-19 pandemic to obtain a proof of concept. Moreover, at the time when the animal feasibility studies were carried out, there were no suitable models, such as the current one with virus-infected animals. Rest assured, in the future, we are going to carry out these studies, but currently, there are no laboratories in Israel that can conduct such an experiment.

In addition, progress in the study of Covid-19 disease in rodents has been hampered by the lack of angiotensin-converting enzyme 2 (ACE2; virus entry route to the target cell) affinities for the virus spike proteins across species. The involvement of LPS (also known

as endotoxin) in the pathology of ARDS has been previously documented. Activation of toll-like receptor-4 (TLR4) with LPS during ARDS induces leukocyte recruitment to the lung, activation of pro-inflammatory cytokine release and consequent lung injury induction, which shares many similarities with SARS. The resulted cytokine storm, pneumonia and coagulopathy are commensurate with moderate and severe Covid-19 disease noted in humans.

LPS-induced ARDS is widely used to study host-response patterns in the pulmonary compartment.

No animal model to date has been demonstrated to be able to reproduce the severe phenotype associated with Covid-19 disease after induction with wild-type SARS-CoV-2. The reason for this may be the inability of wild type SARS-CoV-2 to infect rodent respiratory cells. Additionally, genetically engineered mice expressing human ACE2 do not manifest all the severe symptoms of Covid-19.

In the discussion section (page 28), references have been added.

- *Wang HM, Bodenstein M, Markstaller K. Overview of the pathology of three widely used animal models of acute lung injury. Eur Surg Res. 2008;40(4):305-316.*
- *Rittirsch D, Flierl MA, Day DE, et al. Acute lung injury induced by lipopolysaccharide is independent of complement activation. J Immunol. 2008;180(11):7664-7672.*
- *Bahjat Al- Ani, Asmaa M. ShamsEldeen, Samaa S. Kamar, et al., Clin Exp Pharmacol Physiol. 2022; 49-483-491.*

6. Fig. 6. in this waterfall plot, is there any statistical evaluation showing that survival is increased in the treatment group?

It seems that there is a misunderstanding because on pages 10-11 the statistical evaluation is described, as well as, in the results section (page 22).

7. in the mouse models it seems like at least $10E9$ particles are required to achieve an effect. This was $10E9$ per mouse. $10E9$ was also given to humans. However, the human lung is much bigger than an animal lung. This suggests that this dose would not be sufficient to achieve an effect in humans. Even $10E10$ in humans would be far away and much lower compared to the $10E9$ that were given to mice. Please comment

Obviously, a human lung is much bigger than that of an animal. We had used a dose of 10^9 particles that were required to achieve an effect in vivo, also to humans. It is because of the difference in the route of administration. Rather than receiving exosomes directly through their mouth, mice received it through an inhalation cage, a whole-body exposure chamber. An inhalation cage is an acceptable model, particularly for facilitating animal care and uniform distribution of aerosols in the chamber. On the other hand, there is unknown but significant loss in this cage.

8. Given the different dosing groups in humans, was there dose-dependent effect in the inflammatory parameters. It is not clear which group is displayed here.

No dose-dependent effect was seen, which is similar to the clinical and laboratory indices and tests. The cytokine profile confirmed the reduction in the cytokine storm, but there were no statistically significant differences among the four patient groups.

Sincerely,

Nadir Arber MD, MSc, MHA
Professor of Medicine and Gastroenterology
Yecheil and Helen Lieber Professor for Cancer Research
Chair - Israeli Gastroenterological Association
Head - The Integrated Cancer Prevention Center
Tel-Aviv Sourasky Medical Center
6 Weizman St. Tel Aviv Israel 64239
Tel: 972-3-697-3561/4968 Fax: 972-3-697-4867
Head – Djerassi-Elias Institute ofOncology

8th Jun 2022

Dear Prof. Arber,

Thank you for the submission of your revised manuscript to EMBO Molecular Medicine. I am pleased to inform you that we will be able to accept your manuscript pending the following final amendments:

1) Manuscript format: Please reformat your manuscript according to our guidelines outlined here:

<https://www.embopress.org/page/journal/17574684/authorguide#researcharticleguide>

2) Figures: Please reformat your figures (e.g. reduce contrast of western blot images, reassemble Figure 4C, some figures can be fused together like Figures 6 and 7; 8 and 9) and submit one file per figure. Please note that each figure legend must contain a heading, Figure 10 is currently missing one. Please follow our figure formatting guidelines:

<https://www.embopress.org/page/journal/17574684/authorguide#figureformat>

3) In the main manuscript file, please do the following:

- Correct/answer the track changes suggested by our data editors by working from the attached document.

- Remove all figures and leave figure legends at the end of the manuscript.

- Add callouts for each panel of Figures 3, 5 and 7. Also, figure callouts should be in a sequential order, currently Figure 9B is called out before Figure 9A.

- In M&M, add statistical paragraph that should reflect all information that you have filled in the Authors Checklist, especially regarding randomization, blinding, replication.

- In M&M, please include statement provided in the "Checklist" that the informed consent was obtained from all human subjects and that the experiments conformed to the principles set out in the WMA Declaration of Helsinki and the Department of Health and Human Services Belmont Report.

- Indicate in legends exact $n=$ and exact $p=$ values, not a range, along with the statistical test used. To keep the figures "clear" some authors found providing an Appendix table S_x with all exact p -values preferable. You are welcome to do this if you want to. Please be reminded to show raw values and avoid statistical analyses when $n=2$. Please check our Author Guidelines:

<https://www.embopress.org/page/journal/17574684/authorguide#statisticalanalysis>

- Please rename "Conflict of Interest" to "Disclosure Statement & Competing Interests". We updated our journal's competing interests policy in January 2022 and request authors to consider both actual and perceived competing interests. Please review the policy <https://www.embopress.org/competing-interests> and update your competing interests if necessary.

- Correct the reference citation in the text and reference list. In the text, a reference should be cited by author and year of publication. Include a space between a word and the opening parenthesis of the reference that follows. In the reference list, citations should be listed in alphabetical order. Where there are more than 10 authors on a paper, 10 will be listed, followed by "et al.". Please check "Author Guidelines" for more information.

<https://www.embopress.org/page/journal/17574684/authorguide#referencesformat>

- Provide data availability statement. If no data are deposited in public repositories, please add the sentence: "This study includes no data deposited in external repositories".

Please check "Author Guidelines" for more information.

<https://www.embopress.org/page/journal/17574684/authorguide#availabilityofpublishedmaterial>

4) Author: We noticed that Gil Shenberg is entered as Gil Shainberg in our submission system. Please make sure that the correct author's name is in the manuscript and in our submission system.

5) The Paper Explained: Please provide "The Paper Explained" and add it to the main manuscript text. Please check "Author Guidelines" for more information. <https://www.embopress.org/page/journal/17574684/authorguide#researcharticleguide>

6) Synopsis: Every published paper now includes a 'Synopsis' to further enhance discoverability. Synopses are displayed on the journal webpage and are freely accessible to all readers. They include separate synopsis image and synopsis text.

- Synopsis image: Please provide a striking image or visual abstract as a high-resolution jpeg file 550 px-wide x (250-400)-px high to illustrate your article.

- Synopsis text: Please provide a short standfirst (maximum of 300 characters, including space) as well as 2-5 one sentence bullet points that summarise the paper as a .doc file. Please write the bullet points to summarise the key NEW findings. They should be designed to be complementary to the abstract - i.e. not repeat the same text. We encourage inclusion of key acronyms and quantitative information (maximum of 30 words / bullet point). Please use the passive voice.

7) For more information: Please remove corresponding author's e-mail address. This space should be used to list relevant web links for further consultation by our readers. Could you identify some relevant ones and provide such information as well? Some examples are patient associations, relevant databases, OMIM/proteins/genes links, author's websites, etc...

8) Source data: We encourage you to include the source data for figure panels that show essential data. Numerical data should be provided as individual .xls or .csv files (including a tab describing the data). For blots or microscopy, uncropped images should be submitted (using a zip archive if multiple images need to be supplied for one panel). Please check "Author Guidelines" for more information. <https://www.embopress.org/page/journal/17574684/authorguide#sourcedata>

9) As part of the EMBO Publications transparent editorial process initiative (see our Editorial at

<http://embomolmed.embopress.org/content/2/9/329>), EMBO Molecular Medicine will publish online a Review Process File (RPF) to accompany accepted manuscripts. This file will be published in conjunction with your paper and will include the anonymous referee reports, your point-by-point response and all pertinent correspondence relating to the manuscript. Let us know whether you agree with the publication of the RPF and as here, if you want to remove or not any figures from it prior to publication. Please note that the Authors checklist will be published at the end of the RPF.

10) Please provide a point-by-point letter INCLUDING my comments as well as the reviewer's reports and your detailed responses (as Word file).

I look forward to reading a new revised version of your manuscript as soon as possible.

Yours sincerely,

Zeljko Durdevic

The authors performed the requested editorial changes.

We are pleased to inform you that your manuscript is accepted for publication and is now being sent to our publisher to be included in the next available issue of EMBO Molecular Medicine.